# Identification of a Golgi GPI-N-acetylgalactosamine transferase with tandem transmembrane regions in the catalytic domain

Tetsuya Hirata[1,2,5], Sushil K. Mishra [3], Shota Nakamura[1], Kazunobu Saito[1], Daisuke Motooka[1], Yoko Takada[2], Noriyuki Kanzawa[1,2], Yoshiko Murakami[1,2], Yusuke Maeda[1,2], Morihisa Fujita[4], Yoshiki Yamaguchi[3] & Taroh Kinoshita [1,2]

Many eukaryotic proteins are anchored to the cell surface via the glycolipid glycosylphosphatidylinositol (GPI). Mammalian GPIs have a conserved core but exhibit diverse *N*-acetylgalactosamine (GalNAc) modifications, which are added via a yet unresolved process. Here we identify the Golgi-resident GPI-GalNAc transferase PGAP4 and show by mass spectrometry that PGAP4 knockout cells lose GPI-GalNAc structures. Furthermore, we demonstrate that PGAP4, in contrast to known Golgi glycosyltransferases, is not a single-pass membrane protein but contains three transmembrane domains, including a tandem transmembrane domain insertion into its glycosyltransferase-A fold as indicated by comparative modeling. Mutational analysis reveals a catalytic site, a DXD-like motif for UDP-GalNAc donor binding, and several residues potentially involved in acceptor binding. We suggest that a juxtamembrane region of PGAP4 accommodates various GPI-anchored proteins, presenting their acceptor residue toward the catalytic center. In summary, we present insights into the structure of PGAP4 and elucidate the initial step of GPI-GalNAc biosynthesis.

[1] Research Institute for Microbial Diseases, Osaka University, Suita, Osaka 565-0871, Japan. [2] WPI Immunology Frontier Research Center, Osaka University, Suita, Osaka 565-0871, Japan. [3] Structural Glycobiology Team, Systems Glycobiology Research Group, RIKEN Global Research Cluster, Wako, Saitama 351-0198, Japan. [4] Key Laboratory of Carbohydrate Chemistry and Biotechnology, Ministry of Education, School of Biotechnology, Jiangnan University, Wuxi, Jiangsu 214122, China. [5] Present address: National Institute for Physiological Sciences, National Institutes of Natural Sciences, Okazaki, Aichi 444-8787, Japan. Correspondence and requests for materials should be addressed to T.K. (email: tkinoshi@biken.osaka-u.ac.jp)

Glycosylphosphatidylinositol (GPI) anchoring is a post-translational modification of proteins by a glycolipid, GPI. In mammalian cells, >150 proteins are modified by GPI for anchoring to the cell surface[1]. The common structure of GPI among organisms is EtNP-6Manα1-2Manα1-6Manα1-4GlcNα1-6myo-inositol-phospholipid (where EtNP, Man, and GlcN are ethanolamine phosphate, mannose, and glucosamine, respectively). The 2-position of the first Man linked to GlcN is modified by EtNP in mammalian cells (Fig. 1a). The biosynthesis of GPI occurs in the endoplasmic reticulum (ER) followed by attachment

of the GPI to proteins to generate immature GPI-anchored proteins (GPI-APs). The immature GPI-APs then undergo lipid and glycan remodeling. Post-GPI attachment to proteins 1 (PGAP1) removes the acyl chain linked to the 2-position of inositol and then PGAP5 removes EtNP from the second Man in the ER before the GPI-APs exit from the ER[2–4]. In the Golgi, GPI-APs undergo fatty acid remodeling where PGAP3 removes an sn-2-linked unsaturated fatty acid and PGAP2 is involved in reacylation with stearic acid, a saturated fatty acid[5, 6]. Fatty acid remodeling is crucial for ensuring that GPI-APs associate with

**Fig. 1** Detection of GalNAc side-chain of mammalian free GPIs by T5 mAb. **a** GPI structures in wild-type (WT), Lec8 and 3BT5 CHO cells. − and +, T5 staining negative and positive, respectively. **b** 3B2A (WT) cells (left) and PIGV-mutant CHO cells transiently transfected with empty vector (center) or PIGV cDNA (right) were treated with (bottom) or without (top) pronase and then stained with T5 mAb and polyclonal second antibody. **c** Staining of Lec8 cells with T5 mAb. 3B2A (WT) cells and Lec8 cells transiently transfected with empty vector or SLC35A2 cDNA were stained with T5 mAb and second antibody. **d** Staining of 3BT5 cells transiently transfected with vector or SLC35A2 with T5 mAb and monoclonal second antibody. **e** GPI specificity of T5 mAb. Cells were stained with T5 mAb, anti-uPAR, or FLAER with or without PI-PLC treatment. **b**–**e** Experiments were repeated at least twice. Dotted lines, background staining without first antibody. See also Supplementary Fig. 1

specific membrane domains termed lipid-rafts or lipid-microdomains. If GPI remodeling is disrupted, proteins containing abnormal GPI structures are expressed on the cell surface. Mutations in GPI biosynthetic and remodeling enzymes in humans cause several clinical symptoms, including intellectual disability, epilepsy, and abnormal facial features[7]. This demonstrates clearly that the correct GPI structure is functionally important. Thus understanding the remodeling pathway of GPI is important for revealing the biological significance of GPI anchoring.

In some GPI-APs, GPI can be further modified by glycan side-chains. In mammalian cells, the fourth Man can be attached to the third Man via an α1-2 linkage, by an ER-resident mannosyltransferase PIGZ[8, 9]. N-acetylgalactosamine (GalNAc) can be attached to the first Man via a β1-4 linkage[10]. The GalNAc residue may be elongated by a β1-3 linked galactose (Gal) and the Gal by sialic acid (Sia) (Fig. 1a)[11]. The addition of glycan side-chains gives structural diversity to GPIs. The physiological roles of these side-chains, however, are largely unknown because the enzymes for GalNAc-initiated side-chain addition have remained unknown since the existence of the side-chain was reported in 1988[10]. Recently, a pathological role of the GPI-GalNAc side-chain was suggested. Prions with a desialylated GPI inhibit the expansion of the disease form of prion (PrP$^{sc}$) and also decrease the amount of PrP$^{sc}$[12]. This indicates that sialylation of GPI contributes to prion diseases and that enzymes involved in the addition of the GPI-GalNAc side-chain represent drug targets. However, clarification of the biosynthetic pathway of the GPI-GalNAc side-chain is required to understand the significance of the sialylation of GPI in vivo.

Glycosyltransferases (GTs) play pivotal roles in generating complex glycans[13]. Currently, GTs are classified into 103 families (in the Carbohydrate-Active Enzymes database) based on their amino acid sequence similarity. Interestingly, despite this diversity, the 3D structures of GTs fall into only three structural folds: GT-A, GT-B, and GT-C[14]. Almost all GT-A and GT-B enzymes that are localized in the Golgi exhibit the type II topology with an N-terminal short cytoplasmic peptide, a single transmembrane domain (TMD), a stem region, and the GT fold[15]. These enzymes use nucleotide-sugars as donor substrates. One noteworthy characteristic of most GT-A enzymes is the DXD motif involved in the coordination of a divalent cation that is essential for binding the nucleotide-sugar[14, 16].

In this report, we aim to progress our understanding of the biosynthetic pathway of GPI-GalNAc side-chain. To this end, the identification of a β1-4-GalNAc transferase, termed PGAP4, is presented. PGAP4 is a Golgi-resident GT with the GT-A fold and surprisingly has three TMDs. Results of homology modeling show that the GT-A fold of PGAP4 is split by tandem TMDs, positioning a catalytic cavity near the membrane to accommodate GPI glycolipid. Our findings uncover the initial step of GPI-GalNAc biosynthesis and provide a platform to address the biological significance of this modification. Moreover, the identification of a GT with a unique topology expands the structural repertoire of the Golgi-resident GTs.

## Results

**T5 antibody recognizes mammalian free GPI modified by GalNAc.** Previously, monoclonal antibodies (mAbs) were generated against free GPIs, i.e., non-protein linked GPIs, of the parasite *Toxoplasma gondii*[17]. *T. gondii*-free GPI has the βGalNAc side-chain linked to the 4-position of the first Man and the GalNAc can be modified by glucose (Glc)[18]. One of the antibodies, termed T5-4E10 mAb, hereafter "T5", was reported to detect the GalNAc side-chain lacking Glc[18]. We tested whether

it is useful as a probe for the GPI-GalNAc side-chain in mammalian cells by assessing the immunoreactivity of T5 mAb in Chinese hamster ovary (CHO) cells by flow cytometry. We observed weak but measurable staining with T5 mAb after pronase treatment (Fig. 1b). The staining was not detected in a mutant CHO cell line defective in the *PIGV* gene, one of the GPI biosynthetic genes[19], thereby confirming the specificity of T5 mAb for GPI (Fig. 1b). Importantly, we found that staining with T5 mAb was very high in Lec8 cells[20] even without pronase treatment. Lec8 is a mutant cell line defective in Gal addition to the GalNAc because of a defect in *SLC35A2*, which encodes the UDP-Gal transporter (Fig. 1a, c). Staining with T5 mAb was efficiently eliminated by phosphatidylinositol-phospholipase C (PI-PLC), which specifically cleaves GPI (Fig. 1e). These results showed that T5 mAb specifically recognizes terminally exposed GalNAc linked to free GPIs and that a significant amount of free GPIs bearing the GalNAc side-chain should be present on the CHO cell surface.

**Forward genetic screening of GPI-GalNAc transferase.** To identify the GPI-GalNAc transferase, we took a forward genetic approach in which randomly mutagenized *SLC35A2*-defective CHO cells were screened for T5 mAb staining-negative cells. The Lec8 cell was not suitable for this approach, because its culture always contains some cells that are not stained by T5 (Fig. 1c and Supplementary Fig. 1a). We first established the 3BT5 cell line containing only a negligible fraction of cells that were negative for T5 staining (Supplementary Fig. 1a). 3BT5 had non-functional *SLC35A2*, which was confirmed by the following experimental results: (1) 3BT5 cells were, like Lec8 cells, heavily stained with GSII or HPA lectins (Supplementary Fig. 1b); (2) staining with T5 mAb decreased when SLC35A2 was expressed (Fig. 1d) and (3) a F264S mutation was identified in *SLC35A2* gene (Supplementary Fig. 1c). Finally, T5 staining of 3BT5 cells was abolished by PI-PLC (Fig. 1e). Thus 3BT5 cells were used in the following experiments.

For random mutagenesis, 3BT5 cells were infected by a retrovirus harboring a gene-trapping plasmid[21, 22], followed by screening with T5 mAb. We repeated cell sorting three times to enrich cells that were negative for T5 staining (see Methods section), resulting in the significant enrichment of mutant cells lacking T5 mAb staining (Supplementary Fig. 2a). To determine the insertion sites of gene-trapping vectors, genomic DNA was prepared from mutant cells before and after cell sorting, followed by deep sequencing. We ranked the mapping efficiency by reads per kilobase of exon per million mapped reads (RPKM) scores, which correspond to a density of sequence reads mapped to exons of a gene, and identified that *TMEM246* (known as *C9orf125* in humans), a previously uncharacterized gene, was greatly enriched in cells sorted three times (S3 cells) (Supplementary Fig. 2b). We termed *TMEM246* as *PGAP4* (post-GPI attachment to proteins 4). PGAP4 is widely conserved among species including *Caenorhabditis elegans* (F35C11.4, sequence ID: NP_495738) but surprisingly not in *T. gondii*. According to BioGPS, the mRNA expression of PGAP4 is limited to specific tissue, including the brain and spinal cord in both humans and mice, and the stomach, heart, and testes in mice[23].

**PGAP4 is essential for generating the GPI-GalNAc side-chain.** To examine whether PGAP4 is required for GPI-GalNAc modification, a *PGAP4* knockout (KO) cell line of 3BT5 was generated by CRISPR-Cas9[24, 25]. PGAP4-KO cells barely stained with T5 mAb, indicating that GalNAc-modified free GPIs were lost (Fig. 2a). We then used matrix-assisted laser desorption ionization-time of flight (MALDI-TOF) mass spectrometry to

determine the structure of protein-bound GPI. Because CD59 is known to have GalNAc on GPI[26], we analyzed tagged-CD59 purified from PGAP4-KO cells transfected with PGAP4 cDNA or vector

and wild-type 3B2A cells. A fragment corresponding to GPI without *N*-acetylhexosamine (HexNAc representing GalNAc) ($m/z = 2531$) was obtained from all three cell types, whereas the fragment corresponding to GPI with HexNAc ($m/z = 2734$) was

detected only in PGAP4-transfected KO cells and wild-type cells (Fig. 2b). Liquid chromatography-electrospray ionization-tandem mass spectrometry (LC-ESI-MS/MS) analysis confirmed these structures (Supplementary Fig. 3) and showed that >80% of GPI was modified by GalNAc in wild-type or PGAP4-rescued KO cells, whereas GalNAc modification was hardly detectable in PGAP4-KO cells (Fig. 2c and Supplementary Table 2). Therefore,

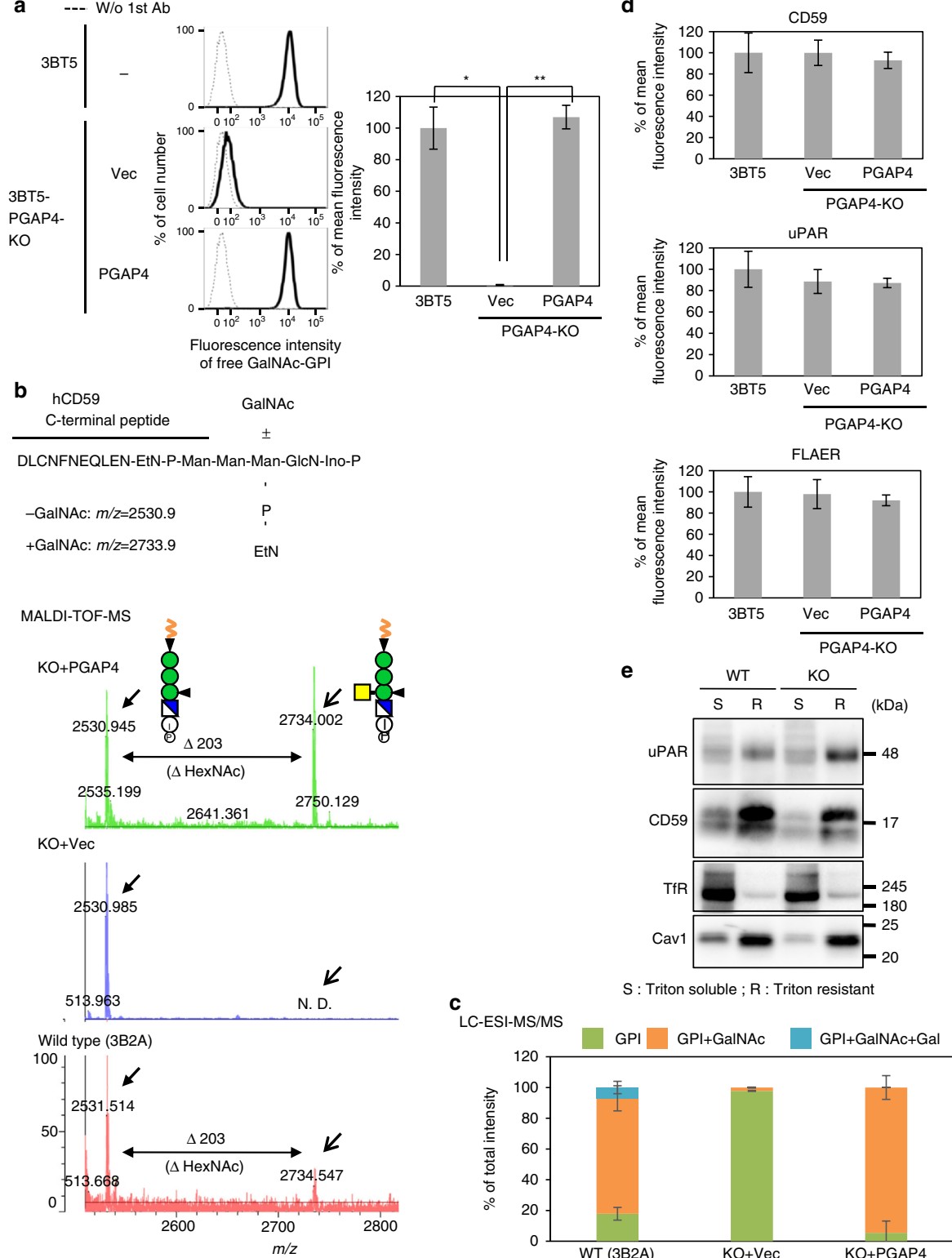

PGAP4 is essential for generating the GalNAc side-chain of both free and protein-bound GPIs.

In PGAP4-KO cells, surface expression of GPI-APs was not altered (Fig. 2d). In addition, the GalNAc side-chain was not involved in raft association of GPI-APs, as GPI-APs were still localized in the detergent-resistant membrane (DRM) in PGAP4-KO cells (Fig. 2e). These results suggest that the GalNAc side-chain does not affect the fundamental nature of GPI.

**PGAP4 is a Golgi-resident GalNAc transferase with three TMDs**. PGAP4 is predicted to be a transmembrane protein with one N-glycosylation site and three TMDs (Fig. 3a and Supplementary Fig. 4a–c). Western blotting with Endo H or PNGase F treatment indicated that PGAP4 has a complex-type N-glycan (Fig. 3b). Immunofluorescent imaging revealed that PGAP4 was colocalized predominantly with GPP130, suggesting that PGAP4 is a Golgi membrane protein (Fig. 3c). DRM separation analysis revealed that PGAP4 is not incorporated into the raft fraction (Fig. 3d).

To examine PGAP4 topology experimentally, we artificially introduced an N-glycosylation site into the PGAP4 amino acid sequence (Fig. 3e, upper). To eliminate the native N-glycosylation site (N87), we first mutated this residue to A (N87A), and this mutant completely lost N-glycosylation (Fig. 3e, lower). Two potential mutants with new N-glycosylation sites were then generated, PGAP4 (N87A, F283N) and (N87A, T347N) (Fig. 3e, upper). T347N was N-glycosylated, whereas F283N was not (Fig. 3e, lower). Although it is known that some potential N-glycosylation sites in the lumen are left unglycosylated, this result together with the computational topology prediction (Supplementary Fig. 4c) suggests that PGAP4 has three TMDs (TM1, TM2, and TM3), and that the region between TM1 and TM2 and the C-terminus face the lumen.

A homology search by PSI-BLAST using the PGAP4 amino acid sequence hit the N-acetylglucosamine (GlcNAc) transferase, GnT-IV (Supplementary Fig. 5a). GnT-IV transfers GlcNAc to the Man of N-glycan via a β1-4 linkage and PGAP4 is required for the addition of GalNAc to Man of GPI via a β1-4 linkage (Supplementary Fig. 5b). Therefore, these two proteins may be expected to have similar structure. We found that GnT-IV and PGAP4s from various species have a conserved EDD motif similar to the DXD motif that is functionally important for many GTs in binding donor substrates (Supplementary Fig. 5a, c). These results support the idea that PGAP4 is a Golgi-resident GalNAc transferase.

**PGAP4 has a GT-A fold split by an insertion of tandem TMDs**. To understand the structure–function relationship of PGAP4, a three-dimensional (3D) structural model of PGAP4 was predicted. Due to a lack of the significant homology with any known 3D structure available in the Protein Data Bank, comparative

modeling of full-length PGAP4 could not be achieved. Instead, multiple threading approach using the PGAP4 amino acid sequence lacking TM2 and TM3 (PGAP4ΔTM2, TM3) identified sequence similarity with bacterial cellulose synthase (Protein Data Bank (PDB) ID: 4P02[27]) (Supplementary Fig. 6). Hence, we first used a fold recognition approach, Iterative Threading ASSEmbly Refinement (I-TASSER)[28], to construct an atomic model of PGAP4ΔTM2, TM3 using the crystal structure of bacterial cellulose synthase (PDB ID: 4P02A) as the template structure. The overall quality of the predicted model was analyzed by PROSA-Web[29] and RAMPAGE[30] as described in the Methods section, which indicated that the model structure is of comparable quality to low-resolution experimental structure of the similar size. The model shows that PGAP4ΔTM2, TM3 consists of a short tail (residues 1–19), the N-terminal TMD (TM1), a luminal helical region (residues 43–82), and a luminal core domain (residues 83–261 and 309–403). The modeled PGAP4ΔTM2, TM3 aligns with the catalytic domain and the preceding helical region of the cellulose synthase (Fig. 4a). The luminal core domain of PGAP4ΔTM2, TM3 consists of a seven-stranded β-sheet surrounded by α-helices, aligning well with the GT-A fold of the cellulose synthase (Fig. 4a). Within the GT-A domain of PGAP4ΔTM2, TM3, the N-terminal part and the C-terminal part closely interact with each other (Fig. 4b). The fold similarities detected by I-TASSER suggest that PGAP4 contains the GT-A fold separated by tandem TMDs. The luminal helical region between TM1 and the GT-A domain corresponds to a stem region in typical Golgi GTs of type II topology (Fig. 4b).

**Validated PGAP4 3D structure with three TMDs**. The PGAP4 sequence contains eight conserved cysteine residues in the luminal regions (Supplementary Fig. 5c), indicating the possibility of disulfide bond formation. An atomic model of the luminal core domain of PGAP4 (Δ1−19, Δ262−308) stipulates for three pairs of cysteine residues, C132–C136, C144–C194, and C332–C333, that are sufficiently close to each other to form disulfide bonds (Cα distances: 10.2, 8.9, and 3.8 Å for C132–C136, C144–C194, and C332–C333, respectively) (Supplementary Fig. 7a). To validate these disulfide pairs, mass spectrometry of purified PGAP4 was conducted. The result confirmed the presence of three disulfide bridges, C132–C136, C144–C194, and C332–C333 (Supplementary Fig. 7b, c). The two remaining cysteines, C43 and C356, are located quite far from each other and cannot interact in the model. Mutational analysis confirmed that C43 and C356 are in free (reduced) form (Supplementary Fig. 7d–g). These lines of experimental evidence are consistent with the model of PGAP4, supporting that the 3D structural model of PGAP4 is valid. Finally, we used the structure of the PGAP4 GT-A fold and the 3D structure of cellulose synthase as templates to model the overall PGAP4 structure with three TMDs using a comparative modeling approach[31]. A series of distance

**Fig. 2** Analysis of PGAP4-KO cells. **a** Left. 3BT5 CHO cells (top) and 3BT5-PGAP4-KO cells stably harboring pLIB2-Hyg (Vec) or pLIB2-Hyg-hPGAP4-3HA (PGAP4) were stained with T5 mAb. Dotted lines, background staining. Right. Mean fluorescence intensity (±SD) from three independent experiments (n = 3). Statistical analyses were done by unpaired Student's t-test. *p < 0.05; **p < 0.005. **b** and **c** A lack of GalNAc side-chain in GPI of CD59 in PGAP4-KO cells. GPI-containing peptide from PI-PLC-treated CD59 was analyzed by MALDI-TOF-MS (**b**) or LC-ESI-MS/MS (**c**). **b** The C-terminal peptide containing GPI after trypsin digestion is shown (top). The m/z of GPI-containing peptides with or without HexNAc are 2733.9 and 2530.9, respectively. **c** Quantification of MS data. Percentage of total intensity (mean ± SD) was calculated from the peak areas obtained by two independent measurements. **d** Normal levels of GPI-APs on PGAP4-KO cells. 3BT5 cells and 3BT5-PGAP4-KO cells stably harboring pLIB2-Hyg (Vec) or pLIB2-Hyg-hPGAP4-3HA (PGAP4) were stained with anti-CD59 (top), anti-uPAR (center) and FLAER (bottom). Mean fluorescence intensity (±SD) from three independent experiments (n = 3). Statistical analyses were done by unpaired Student's t-test. Not significant. **e** DRM separation experiment. Transferrin receptor (TfR) and Caveolin1 (Cav1) are markers of Triton-soluble (S) and -resistant (R) fractions, respectively. Representative data from two independent experiments. See also Supplementary Fig. 2, 3

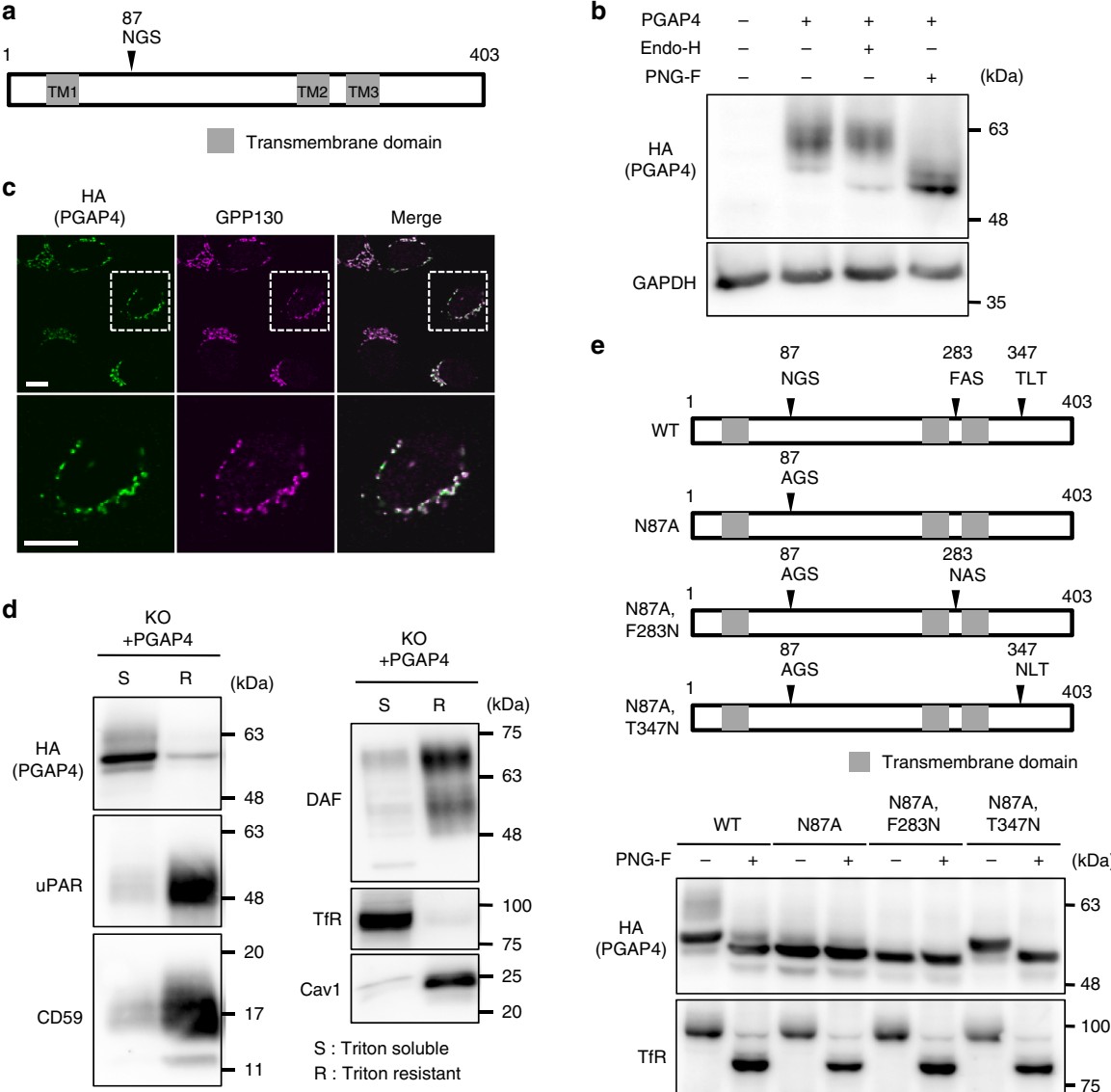

**Fig. 3** Characterization of PGAP4. **a** Schematic of PGAP4 protein. Arrowhead, an *N*-glycosylation site; gray boxes, transmembrane domains. **b** Western blotting of PGAP4. Lysates of 3B2A cells stably expressing PGAP4-3HA were treated with Endo H or PNGase F followed by western blotting. GAPDH, a loading control. **c** Immunofluorescence imaging of PGAP4. 3B2A was transiently transfected with pME-hPGAP4-3HA. Lower panels are enlarged images of regions surrounded by white squares in the upper panels. GPP130, the Golgi marker. Scale bars: 10 μm. **d** Non-DRM localization of PGAP4. Lysate of 3BT5-PGAP4-KO cells transiently expressing PGAP4-3HA were fractionated into Triton-soluble (S) and -resistant (R) fractions and subjected to western blotting. **e** Analysis of PGAP4 topology. (Upper) Schematic of PGAP4 WT and *N*-glycosylation mutants. Arrowheads, target sites of mutations. (Lower) Western blotting of PGAP4 WT and mutants. pME-hPGAP4-3HA was transiently transfected in 3B2A cells. Lysates were treated with or without PNGase F. TfR, a positive control for PNGase F treatment. **b–e** Representative data from two independent experiments. See also Supplementary Fig. 4 and 5

restraints was used to assemble three TMDs in a plane and to incorporate the disulfide bonds in the model (see Methods section), generating a complete 3D structural model of PGAP4 (Fig. 4c, Supplementary Fig. 8, Supplementary Data 1, and Supplementary Movie 1).

**Characterization of the catalytic region of PGAP4.** Structural superposition of the GT-A fold in PGAP4 and the cellulose synthase indicates a putative cavity accommodating a donor substrate UDP-GalNAc (Fig. 4d). The DXD motif (D246-A247-D248) of the cellulose synthase, which coordinates an $Mg^{2+}$ ion for UDP-Glc binding, is superimposable with the E211-D212-D213 of PGAP4. To determine the functional significance of this motif, PGAP4 mutants E211A and D213A were constructed. As

expected, both mutants completely lost the activity to restore GalNAc side-chain in 3BT5-PGAP4-KO cells (Fig. 4e–g). These results demonstrate that E211-D212-D213, a DXD-like motif, is essential for the activity of PGAP4.

The catalytic center of the cellulose synthase, D343, which is close to the DXD motif, corresponds to the D363 in PGAP4, which is conserved among species (Fig. 4d and Supplementary Fig. 5c). A D363A mutant of PGAP4 also lost the activity (Fig. 4h–j), indicating D363 to be a putative catalytic center of this enzyme.

**Mutational analyses highlight the UDP-GalNAc-binding site.** To simulate the binding mode of PGAP4 to a donor substrate, we incorporated UDP-GalNAc into the model by adopting UDP-Glc from the cellulose synthase structure during the homology

modeling (Fig. 5a). Residue D363 is located near the GalNAc where it can mediate catalysis (Fig. 5a, b and Supplementary Movie 1). The model shows that V109, T334, P335, and K362, all of which are conserved residues, form a cavity for UDP-GalNAc (Fig. 5b and Supplementary Fig. 5c). Activities of V109A, P335A, and K362A mutants were reduced significantly, and the T334A mutant was almost inactive; all these mutants showed the proper Golgi localization and wild-type-level expression (Fig. 5c–e). These results indicate that all four residues form a cavity that accommodates UDP-GalNAc.

**PGAP4 may interact with GPI at a juxtamembrane region.** The PGAP4 model demonstrated that the enzyme has a concave surface in the juxtamembrane region near the catalytic site (Fig. 6a and Supplementary Movie 1). The open space between this surface and the membrane seems sufficiently wide to accommodate the GPI glycan (Fig. 5a). To investigate the functional significance of the concave surface, mutational analysis was performed on six residues, H247, E249, M260, H311, F313, and R317, located on this surface (Fig. 6 and Supplementary Fig. 5c and Supplementary Movie 1). The activities of the H247A and

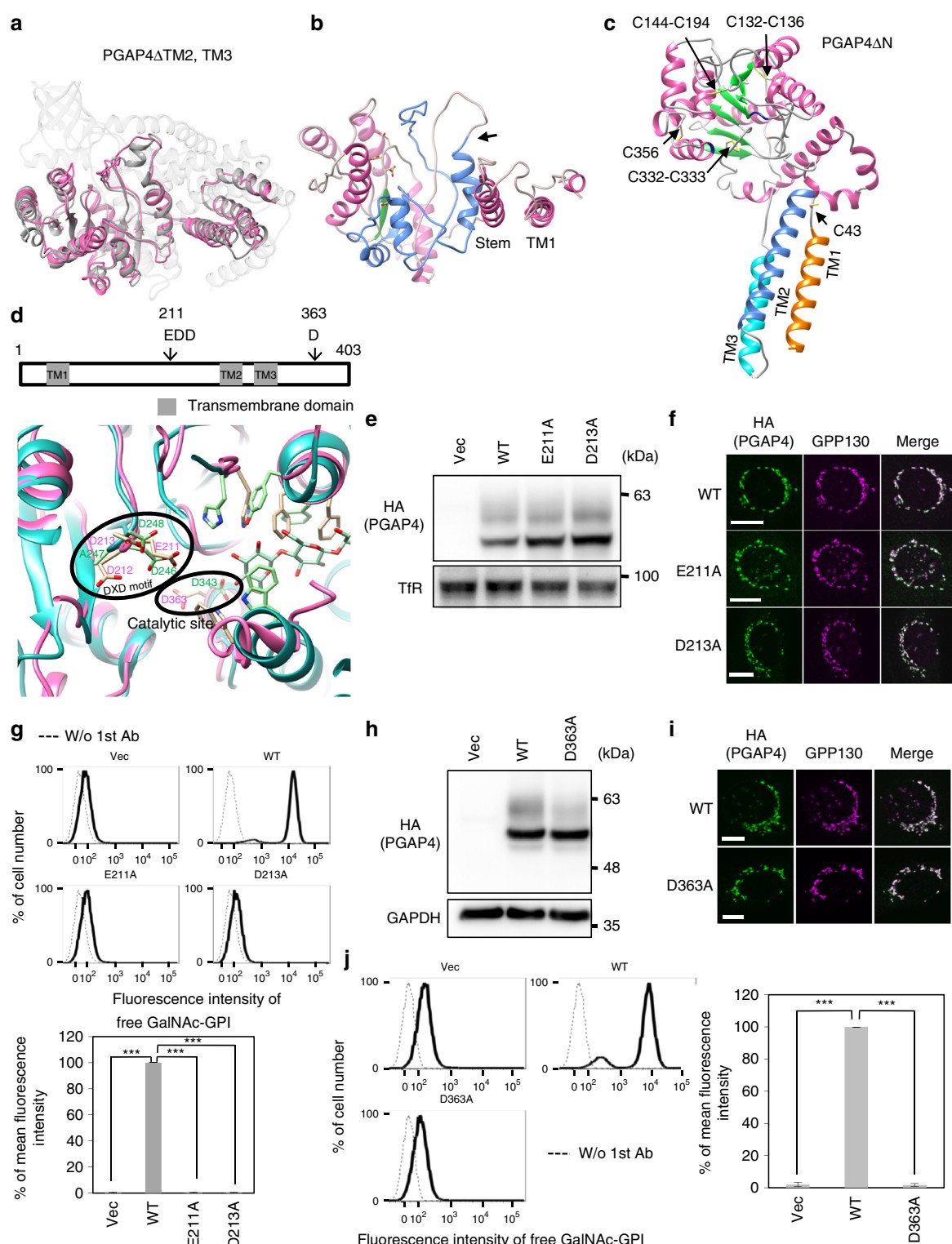

F313A PGAP4 mutants were significantly reduced without affecting the expression or localization (Fig. 6b–d). E249A mutation, but not H311A and M260A, weakly but significantly reduced the activity (Fig. 5c–e and Supplementary Fig. 9). R317A mutation severely affected the activity due to accumulation of this mutant in the ER (Supplementary Fig. 9). Combined with the knowledge gained from the structural model, these results suggest that the concave surface in the juxtamembrane region of PGAP4 is involved in the recognition of the GPI glycan (Fig. 6e). Mutations in TM2 (M270A) and TM3 (M302A) did not alter the enzyme activity (Fig. 5c, e and Supplementary Fig. 9). However, because the TMDs are adjacent to the concave surface, we propose that PGAP4 interacts with the GPI lipid via one or more of the three TMDs (Fig. 6e).

**The N-terminal TMD of PGAP4 is the Golgi targeting signal**. The N-terminal TMD of many Golgi-resident GTs acts as a Golgi-targeting signal[32]. To determine whether TM1 of PGAP4 acts as a Golgi-targeting signal, both the N-terminal cytoplasmic domain and TM1 were deleted (PGAP4ΔTM1). We found that PGAP4ΔTM1 accumulated in the ER (Fig. 6f). We then made a green fluorescent protein (GFP) fusion with TM1 of PGAP4 (TM1-GFP). TM1-GFP colocalized with GPP130 (Fig. 6g). These results indicate that TM1 of PGAP4 functions as a signal for Golgi targeting, as seen for other GTs.

**Substrate specificity and timing of GalNAc transfer**. To investigate the acceptor substrate specificity of PGAP4, we used mutant CHO cell lines each defective in one of the GPI-remodeling enzymes. The cells were also made defective in SLC35A2 to prevent Gal addition to GalNAc, which interferes with probing with T5 mAb. We analyzed the GPI anchor of CD59 by mass spectrometry and/or free GPIs by T5 staining.

EtNP on the first Man is attached by the enzyme PIGN and therefore loss of PIGN activity leads to the loss of EtNP on the first Man (Fig. 7a)[33]. The PIGN-SLC35A2-defective cells partially lost the surface expression of CD59 as expected (Fig. 7b)[33]. However, staining with T5 mAb was comparable to that of wild-type cells (Fig. 7b). Overexpression of PIGN in the PIGN-SLC35A2-defective cells normalized the CD59 expression, but it did not enhance the T5 mAb staining (Supplementary Fig. 10a). These results indicate that the EtNP side-chain on the first Man is dispensable for recognition by PGAP4 and that it is not part of the epitope recognized by the T5 mAb.

PGAP5 removes EtNP from the second Man before GPI-APs exit the ER. Therefore, PGAP5-defective cells have three EtNPs in GPI-APs, whereas wild-type cells have two EtNPs (Fig. 7a)[3]. PGAP5-SLC35A2-defective cells (C19-SLC35A2-KO) were generated accurately as indicated by a similar lectin staining pattern to that of Lec8 cell lines (Supplementary Fig. 10b). Mass spectrometry revealed that the amount of GalNAc modification of CD59 in PGAP5-SLC35A2-defective cells was not reduced (Fig. 7c, Supplementary Fig. 11, and Supplementary Table 3). These results indicate that removal of the EtNP from the second Man is not required for substrate recognition by PGAP4.

GPI-APs undergo fatty acid remodeling in the Golgi (Fig. 7e)[5, 6]. We asked whether GalNAc transfer by PGAP4 occurs before or after fatty acid remodeling. To answer this question, we generated a PGAP2-KO line from 3BT5 cells. PGAP2-defective cells produce GPI-APs bearing the lyso-form of GPI and transport these GPI-APs on the cell surface (Fig. 7a). Proteins bearing the lyso-form GPI have only one hydrocarbon chain for their membrane insertion and are released from the membrane spontaneously or after cleavage by GPI phospholipase D[6]. If GalNAc transfer occurs before the fatty acid remodeling, the fraction of GalNAc-modified GPI may be similar in GPI-APs in wild-type cells and those released into the medium from PGAP2-KO cells. If GalNAc transfer occurs after fatty acid remodeling, a fraction of GalNAc-modified GPI might be smaller in proteins released from PGAP2-KO cells because of altered GPI lipid structure. Experimentally, PGAP2-KO cells showed dramatically reduced surface expression of GPI-APs and T5 mAb staining, which were restored by transfection of PGAP2 cDNA, confirming the successful KO (Supplementary Fig. 10c). Mass spectrometry revealed that the efficiency of GalNAc modification was greatly decreased in PGAP2-KO cells (GPI + GalNAc; 95.5% ± 0.68% [WT] vs. 74.1% ± 0.13% [PGAP2-KO]) (Fig. 7d, Supplementary Fig. 12, and Supplementary Table 4). Thus we concluded that PGAP4 functions after fatty acid remodeling (Fig. 7e).

Finally, we used PGAP3-KO cells to test whether the fine structure of the *sn*-2-linked fatty acyl chain in GPI is critical for recognition by PGAP4. Whereas wild-type cells mostly have a saturated fatty acid (stearic acid) at the *sn*-2 position, PGAP3-KO cells have an unsaturated fatty acid (such as arachidonic acid or docosatetraenoic acid) at the *sn*-2 position, and because of this abnormality the GPI-APs do not efficiently associate with the DRM[5, 6]. We generated a PGAP3-KO line from 3BT5 cells. The DRM separation experiment confirmed the successful KO of

**Fig. 4** 3D structural modeling of PGAP4 demonstrating the unique split GT-A fold. **a** 3D structural model of PGAP4 lacking two TMDs (TM2 and TM3, corresponding to 262–308). The structure of PGAP4ΔTM2, TM3 (pink) is superimposed to GT domain of the crystal structure of cellulose synthase (dark gray, PDB: 4P02A). Non-GT regions of cellulose synthase are in light gray. **b** The structural model of PGAP4ΔTM2, TM3 exhibited split GT-A fold. Pink region, a region N-terminal to TM2; blue region, a region C-terminal to TM3; arrow, the joint after removal of two TMDs. **c** 3D structural model of PGAP4 with disulfide bridges and three TMDs displayed in ribbon model. PGAP4ΔN (Δ1–19) is shown. Cysteine residues are displayed with sticks. **d** Enlarged view of superimposed catalytic regions. (Upper) Schematic of PGAP4. Arrows, DXD-like motif and putative catalytic site. (Lower) Functionally important residues are shown with sticks. Pink: PGAP4ΔTM2, TM3; cyan: cellulose synthase (PDB: 4P02A). Cellulose is displayed. **e–g** Analysis of DXD-like motif mutants of PGAP4 by western blotting (**e**), immunofluorescence imaging (**f**), and flow cytometry (**g**). pME-hPGAP4-3HA-bearing WT or mutant sequences was transiently transfected into 3BT5-PGAP4-KO cells and analyzed. **e** TfR, loading control. **g** Activities of PGAP4 mutants to restore GalNAc-modified GPI as assessed by staining with T5 mAb. (Upper) Dotted lines, background staining. (Lower) Quantitative data. Statistical analyses were done by unpaired Student's *t*-test. ***$p < 0.0005$. **h–j** Analysis of catalytic mutant of PGAP4 by western blotting (**h**), immunofluorescence imaging (**i**), and flow cytometry (**j**). pME-hPGAP4-3HA-bearing WT or D363A sequence was transiently transfected into 3BT5-PGAP4-KO cells. **h** GAPDH, loading control. **i** GPP130, the Golgi marker. Scale bars: 10 μm. **j** Functional analysis of the catalytic site mutant. Cells were stained with T5 mAb. (Left) Dotted lines, background staining. (Right) Quantitative data. Statistical analyses were done by unpaired Student's *t*-test. ***$p < 0.0005$. **e**, **f**, **h**, **i** Representative data from two independent experiments. **g**, **j** Mean fluorescence intensity (±SD) from three independent experiments ($n = 3$). T5 staining of cells restored by WT PGAP4 was set as 100%. See also Supplementary Fig. 5, 6, 7 and 8

PGAP3 (Supplementary Fig. 10d). Flow cytometry revealed that T5 mAb staining decreased by 50% in the PGAP3-KO cells with no change in the expression of GPI-APs (Fig. 7f). However, mass spectrometry of protein-bound GPI showed a smaller difference in the efficiency of GalNAc modification (93.5% [WT] vs. 86.2% [PGAP3-KO]) (Fig. 7g and Supplementary Table 5). These results suggest that the fine structure of the sn-2-linked acyl chain is not strictly recognized by PGAP4.

## Discussion

We have characterized the first step in the biosynthetic pathway of the GalNAc side-chain of mammalian GPI. We used T5 mAb, which was developed against a side-chain of free GPIs from T. gondii[17, 18], as a probe for the GPI-GalNAc side-chain. T5 mAb requires terminally exposed GalNAc for its recognition of free GPIs. Although wild-type CHO cells are T5 staining negative, pronase treatment makes them T5 staining positive. Pronase

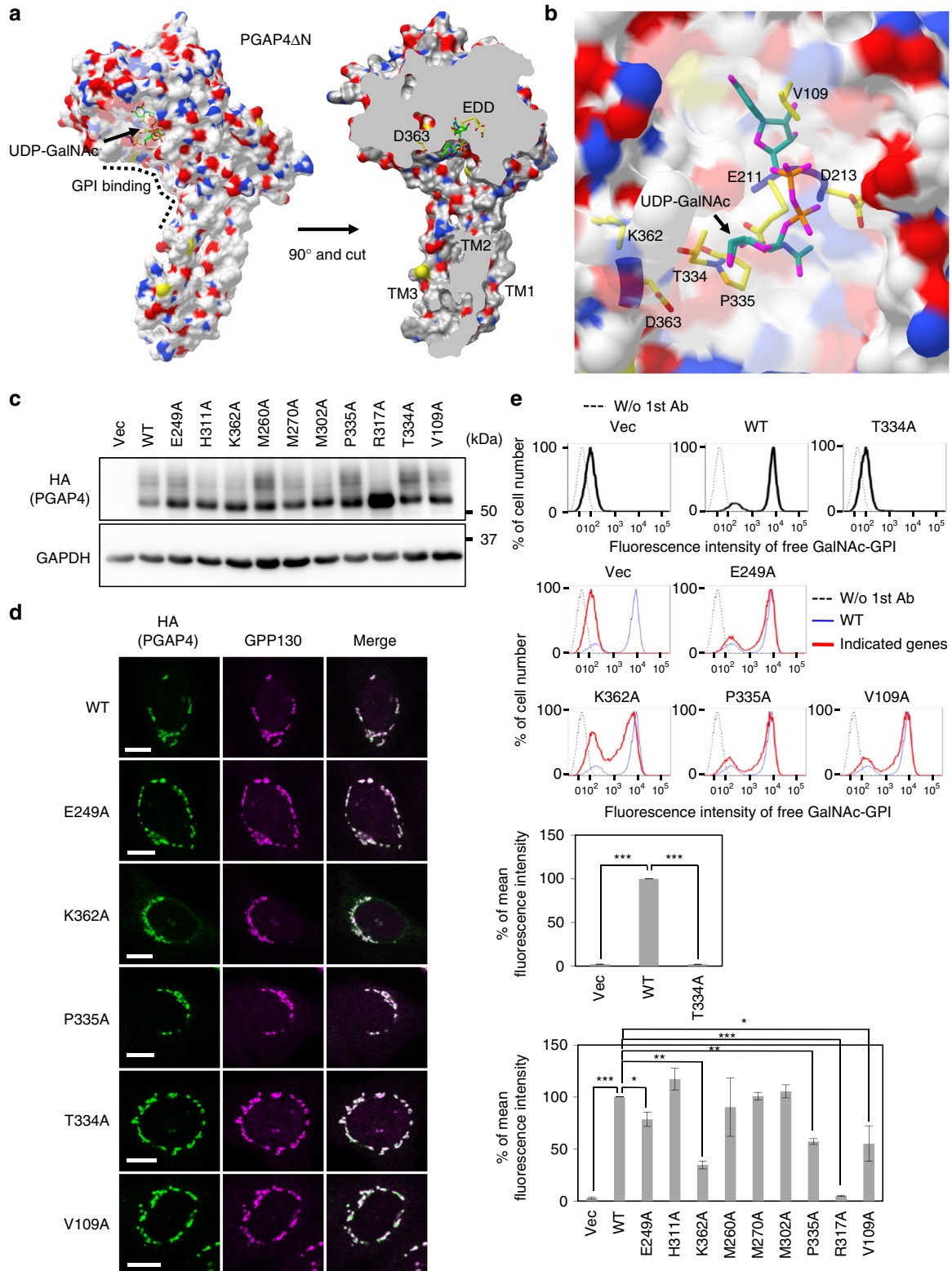

probably cleaves amide bonds between C-terminal residues and EtNP of GPIs[34], thereby free GPIs are newly generated, suggesting that some GPI-APs have terminally exposed GalNAc side-chain. On the other hand, CHO Lec8 cells that are defective in UDP-Gal transport into the Golgi are clearly positive for T5 staining. CHO cells, therefore, have some free GPI, almost all of which is further modified with Gal.

We identified *PGAP4* (originally termed *TMEM246*) as a GPI-modifying GalNAc transferase in the Golgi. Our data demonstrate that GalNAc addition to GPI is carried out in the Golgi after fatty acid remodeling (Fig. 7e). This is consistent with the fact that GalNAc modification of various glycans occurs in the *cis* to medial Golgi[35], whereas fatty acid remodeling may take place in the *cis* Golgi[5, 6]. A lack of PGAP4 in CHO cells did not alter the fundamental nature of the GPI-APs, including surface expression levels and raft association (Fig. 2d, e). So, the biological role(s) of the GalNAc side-chain remain to be clarified.

Since typical Golgi-resident GTs are type II membrane proteins that have a short N-terminal peptide, one TMD, a luminal stem region, and a GT fold[15], PGAP4 shows a unique modular architecture. 3D homology modeling of PGAP4 demonstrated that the structural arrangement of PGAP4 has a short N-terminal peptide, one TMD, a luminal stem region, and the GT-A fold with inserted tandem TMDs. This indicates that PGAP4 shares common structural properties with previously characterized Golgi-resident GTs, aside from the two TMD insertions. Consistent with this, the first TMD of PGAP4 functions as the Golgi-targeting signal similar to other Golgi-resident GTs[32] (Fig. 6f, g). Thus PGAP4 shows both unique and common features found in GTs. PGAP4 is a Golgi-resident GT that shows a split GT-A fold with multiple TMDs. Whether such structural topology exists for other GTs remains to be seen.

We have identified residues critical for enzymatic activity of PGAP4, including a DXD-like motif, the catalytic site, and the region that potentially recognizes GPI glycan. These residues are closely localized to each other, suggesting that reaction can be carried out at their interface. The three TMDs and the functionally essential residues are also close. Mass spectrometry in PGAP3-KO and PGAP2-KO cells indicates that PGAP4 does not distinguish the precise lipid structure of GPI but instead needs two fatty acid chains. Given the 3D structural model, we propose two significances of tandem TMD insertion: (1) to approximate the catalytic site to GPI at the vicinity of the membrane and (2) to interact with GPI lipid by at least one TMD. The unique topology of PGAP4 is suitable to accommodate the glycolipid substrate, GPI (Fig. 6e).

Using PSI-BLAST, PGAP4 orthologs were identified in various species but surprisingly not in *T. gondii*, even though that parasite has GPI with a GalNAc side-chain. A previous study showed that GalNAc modification was carried out in the ER in *T. gondii*[36]. These observations suggest that *T. gondii* uses a GalNAc transferase that is not orthologous to PGAP4. Identification of another GalNAc transferase is thus an aim of future study.

Previous studies revealed that not all GPI-APs are modified with a GalNAc side-chain. This indicates two possibilities. One is limited distribution of PGAP4 expression. The GalNAc side-chain was detected in some GPI-APs purified from brain, kidney and skeletal muscle[10, 11, 37, 38] but not from erythrocytes or placenta[39, 40], consistent with the expression profile of PGAP4 reported in BioGPS. The other possibility is a structural constraint in PGAP4. Although GPI should be sufficient for the recognition by PGAP4 because free GPIs can be modified by GalNAc, some GPI-APs might escape from PGAP4. Based on our structural model, PGAP4 and the C-terminus of the target proteins could be in close proximity. Thus, 3D structure of the target proteins or some modifications to the C-terminal region, such as heavy glycosylation, may interfere with PGAP4 interaction. Further studies are needed to clarify this possible functional feature.

In conclusion, we have uncovered the initial step of the GalNAc modification of GPI. Our findings establish a genetic basis for studying the significance of the GPI-GalNAc side-chain. With this in mind, we are currently preparing PGAP4-KO mice, which will facilitate examination of the physiological and pathological significance of the GPI-GalNAc side-chain in vivo. Moreover, we have provided evidence that PGAP4 is a Golgi-resident GT with an unusual modular arrangement. This may provide a basis for identifying other enzymes with similar architectures, which were not considered to be GTs or remained uncharacterized so far.

## Methods

**Reagents and antibodies.** Following reagents were purchased: Pronase (Roche); FLAER-AlexaFluor488 (Cedarlane); GSII-AlexaFluor647 and HPA-AlexaFluor647 (L32451 and L32454, Thermo Fisher); MAM-fluorescein isothiocyanate (FITC) (J510, J-OIL-MILLS); PI-PLC (P6466, Thermo Fisher); Imperial Protein Stain (24615, Thermo Fisher); RNeasy Mini Kit (74104, QIAGEN); SuperScript VILO Master Mix (11755050, Thermo Fisher); Complete, EDTA-free Protease Inhibitor Cocktail (11873580001, Roche); FLAG-M2 beads and 3× FLAG peptide (A2220 and F4799, Sigma-Aldrich); Glutathione Sepharose 4B (17075601, GE Healthcare); and glutathione (Reduced form, 073–02013, Wako). T5-4E10 (T5) was kindly gifted from Dr. J. F. Dubremetz (Montpellier University[17]). Mouse monoclonal anti-human CD59 (clone 5H8[41]), anti-human DAF (clone IA10, BD, cat# 565939), and anti-hamster uPAR (clone 5D6[41]) were used. Antibodies for human CD59 and hamster uPAR are available from the authors upon request. Following antibodies were purchased: Mouse monoclonal anti-HA (clone HA7, Sigma-Aldrich, cat# H3663), anti-Caveolin 1 (clone 2297, BD, cat# 610407), anti-TfR (clone H68.4, Thermo Fisher, cat# 136800) and anti-GAPDH (clone 6C5, Thermo Fisher, cat# AM4300); rabbit polyclonal anti-GPP130 (Covance, cat# AM4300) and anti-BiP (Affinity BioReagents, cat# PA1-014); rat monoclonal anti-mouse IgM-allophycocyanin (APC) (clone RMM-1, BioLegend, cat# 406509); goat polyclonal anti-mouse IgM-FITC (Thermo Fisher, cat# 31992), anti-mouse IgG-AlexaFluor488 (Thermo Fisher, cat# A11029), anti-rabbit IgG-AlexaFluor594 (Thermo Fisher, cat# A11037) and anti-mouse IgG-phycoerythrin (PE) (BioLegend, cat# 405307); and sheep polyclonal anti-mouse IgG-HRP (GE Healthcare, cat# NA9310-1ML).

**Plasmids.** Primers used here are listed and numbered in Supplementary Table 1. To make the expression plasmid of PGAP4, we amplified hPGAP4 (TMEM246 or C9orf125) from cDNA library made from Hep3B by using primers No. 15 and 16. The amplified DNA was cut with *Eco*RI and *Not*I and cloned into pME-Zeo plasmid cut by the same enzymes. To construct the PGAP4-3HA plasmid, PGAP4 was amplified from pME-Zeo-hPGAP4 by primers No. 15 and 17. The amplified

**Fig. 5** Functionally important residues in the UDP-GalNAc-binding cavity. **a** 3D structural model of PGAP4 with three TMDs in molecular surface display method. (Left) The whole structure of PGAP4ΔN with UDP-GalNAc. (Right) The model rotated by 90° and cut to visualize the active site. Carbon, nitrogen, oxygen, sulfur, and hydrogen are shown in gray, blue, red, yellow, and white, respectively. **b** Enlarged view of PGAP4ΔN around UDP-GalNAc. Residues important for the activity are shown by sticks. **c, d** Western blotting (**c**) and immunofluorescence imaging (**d**) of various PGAP4 mutants. pME-hPGAP4-3HA-bearing WT or mutant sequences was transiently transfected into 3BT5-PGAP4-KO cells. **c** Representative data from two independent experiments. GAPDH, loading control. **d** GPP130, the Golgi marker. Scale bars: 10 μm. **e** Functional analysis of PGAP4 mutants. 3BT5-PGAP4-KO cells were transiently transfected with pME-hPGAP4-3HA for T334A mutant or pTK-hPGAP4-3HA harboring weak promoter for other mutants. Cells were stained with T5 mAb. Dotted lines, background staining. In lower panels, blue and red lines indicate PGAP4 WT and mutants, respectively. Quantitative data, mean fluorescence intensity (±SD) from three independent experiments ($n = 3$). *$p < 0.05$; **$p < 0.005$; ***$p < 0.0005$. **e** Statistical analyses were done by unpaired Student's *t*-test. See also Supplementary Fig. 5 and 9

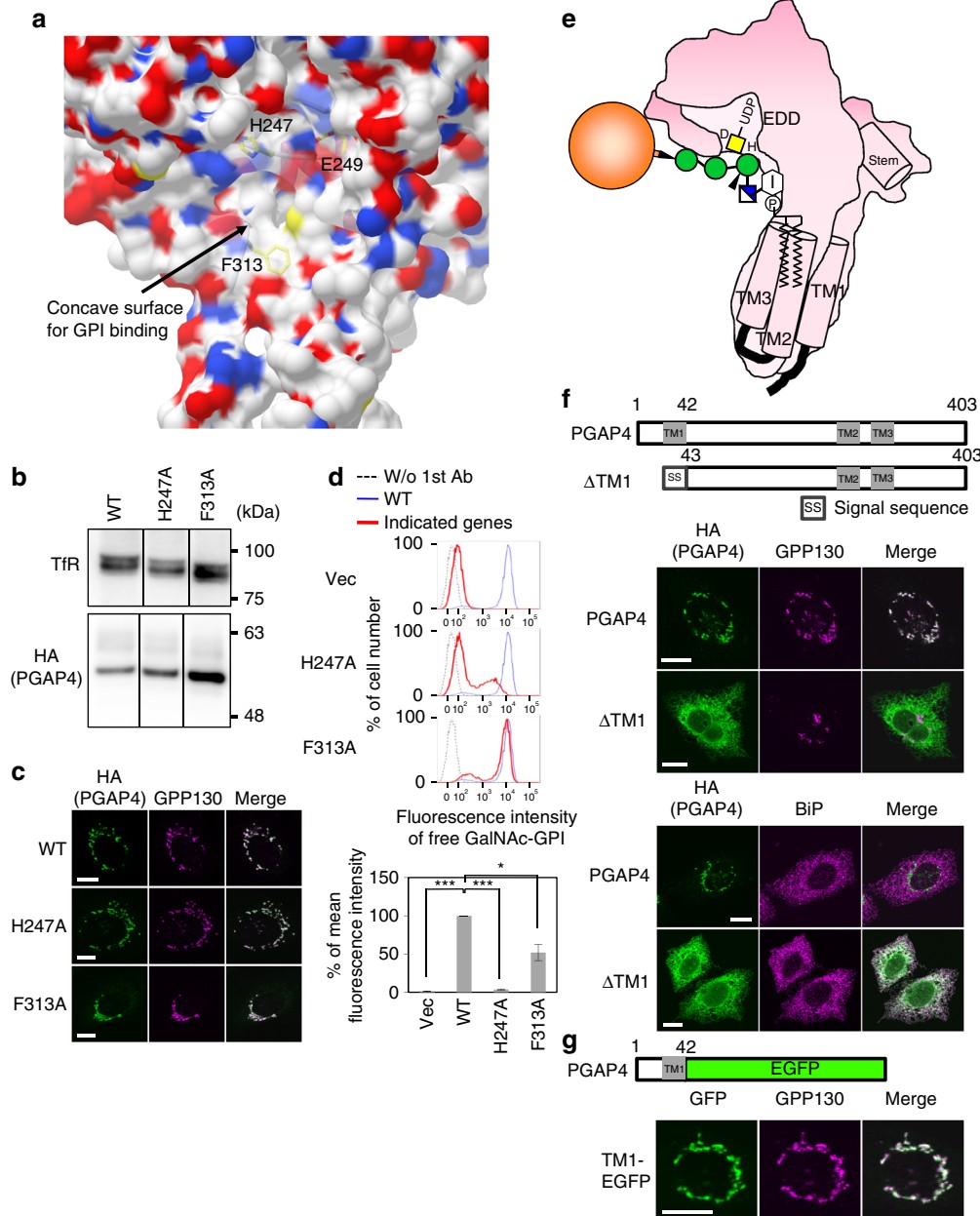

**Fig. 6** Functional analysis of the concave surface in the juxtamembrane region and the first TMD. **a** Enlarged view of the concaving surface near the catalytic cavity. H247, E249, and F313 are exposed on the surface and located between UDP-GalNAc-binding site and TMDs. **b**–**d** Functional analysis of H247A and F313A mutant PGAP4. For western blotting (**b**) and immunofluorescence imaging (**c**), pME-hPGAP4-3HA with WT or mutant sequences was transiently transfected into 3BT5-PGAP4-KO cells. **b** TfR, loading control. **c**, **d** Representative data from two independent experiments. **c** GPP130, the Golgi marker. Scale bars: 10 μm. **d** Flow cytometry. 3BT5-PGAP4-KO cells were transiently transfected with pTK-hPGAP4-3HA. Cells were stained with T5 mAb. Dotted lines, background staining; blue lines, PGAP4 WT; red lines, PGAP4 mutants. Quantitative data of mean fluorescence intensity (±SD) from three independent experiments ($n = 3$). Statistical analyses were done by unpaired Student's $t$-test. *$p < 0.05$; ***$p < 0.0005$. **e** Schematic model for the interaction of PGAP4 with GPI-APs. PGAP4 exhibits a structure like a "golf club head with a short shaft" in which head portion corresponds to the catalytic domain. The catalytic domain harbors the cavity for UDP-GalNAc binding and the GPI-glycan binding concaving surface. The latter is closely located to the shaft corresponding to three bundled TMDs. The first mannose that accepts GalNAc (yellow square) should be presented close to the catalytic site (D363). GPI lipid might interact with TMDs. The space between the head portion and the membrane (not shown) may be sufficiently wide to accommodate entire GPI glycan. The protein part of various GPI-APs may be positioned out of this space not to interfere with the interaction with PGAP4. EDD, DXD-like motif; D, putative catalytic site D363; H, H247 in GPI-binding region. **f** The localization of PGAP4ΔTM1. (Upper) Schematic of the deletion mutant. Gray and white boxes indicated TMDs and signal sequence, respectively. (Lower) Immunofluorescence imaging of PGAP4ΔTM1. 3BT5-PGAP4-KO cells were transiently transfected with pME-hPGAP4-3HA with WT or deletion mutant sequence. GPP130 and BiP, Golgi and ER markers, respectively. Scale bars: 10 μm. **g** Localization of TM1-EGFP. (Upper) Schematic of the chimera. (Lower) 3BT5-PGAP4-KO cells were transiently transfected with pME-TM1-EGFP. GPP130, the Golgi marker. Scale bars: 10 μm. See also Supplementary Fig. 5 and 9

sequence was cut with *Eco*RI and *Mlu*I followed by ligation into pME-3HA plasmid cut by the same enzymes. To construct pLIB2-Hyg-hPGAP4 and pLIB2-Hyg-hPGAP4-3HA plasmid, pME-Zeo-hPGAP4 and pME-hPGAP4-3HA were digested by *Eco*RI and *Not*I followed by ligation with pLIB2-Hyg cut by the same enzymes. To make pME-3FLAG-hPGAP4-3HA plasmid, hPGAP4-3HA was amplified by using primers No. 18 and 19. The amplified sequence was cut with *Sal*I and *Not*I, and ligated into pME-3FLAG plasmid cut by the same enzymes. To make pME-Hyg-3FLAG-hPGAP4-3HA plasmid, 3FLAG-hPGAP4-3HA was cut with *Eco*RI and *Not*I followed by ligation with pME-Hyg plasmid cut by the same enzymes. To make PGAP4ΔTM1 plasmid encoding 1–25 amino acids of hCD59 with 43–403 amino acids of hPGAP4, corresponding sequence was amplified from pME-

hPGAP4-3HA plasmid by using primers No. 20 and 21. The amplified sequence was inserted into pME plasmid with signal sequence of CD59, which was cut by *Pst*I and *Not*I, by the In-Fusion Cloning Kit (TaKaRa). To make TM1-EGFP plasmid encoding 1–42 amino acids of hPGAP4 with enhanced green fluorescent protein (EGFP), corresponding sequence was amplified from pME-hPGAP4-3HA plasmid by using primers No. 15 and 22. The amplified sequence was cut with *Eco*RI and *Mlu*I followed by ligation into pME-mEGFP plasmid cut by the same enzymes. To introduce mutation to plasmids, we followed the QuickChange Site-Directed Mutagenesis method. pME-hPGAP4-3HA was used as a template for the following mutants; N87A (primers No. 23 and 24), E211A (No. 29 and 30), D213A (No. 31 and 32), C43S (No. 35 and 36), C356S (No. 37 and 38), D363A (No. 39 and

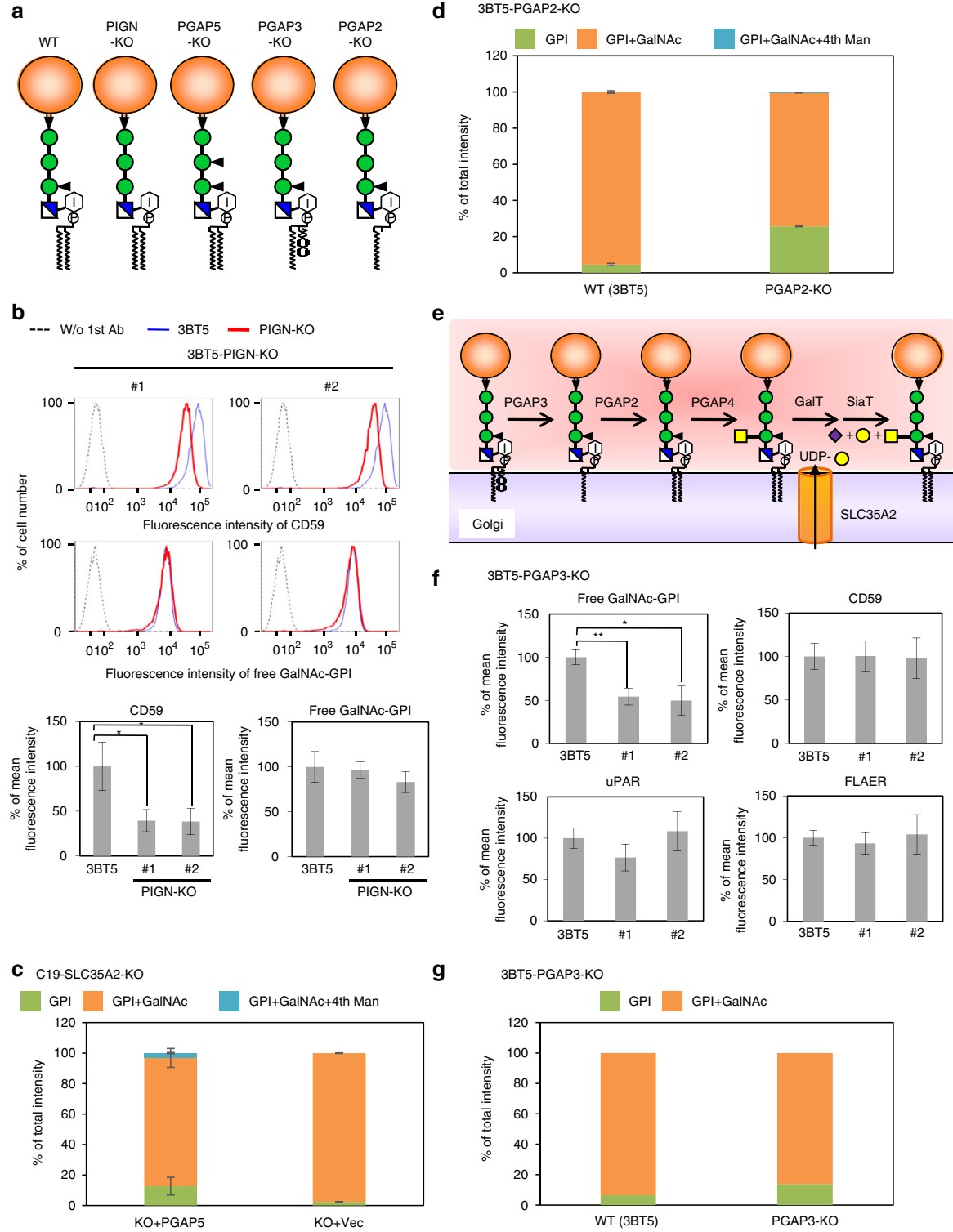

40), E249A (No. 41 and 42), H311A (No. 43 and 44), K362A (No. 45 and 46), M260A (No. 47 and 48), M270A (No. 49 and 50), M302A (No. 51 and 52), P335A (No. 53 and 54), R317A (No. 55 and 56), T334A (No. 57 and 58), V109A (No. 59 and 60), H247A (No. 61 and 62), and F313A (No. 63 and 64). pME-hPGAP4 (N87A)-3HA was used as a template for the following mutants; N87A, F283N (No. 25 and 26), and N87A, T347N (No. 27 and 28). pME-3FLAG-hPGAP4-3HA was used as a template for the mutant Q26K (Primers No. 33 and 34). pME-hPGAP4 (C43S)-3HA was used as a template for the double mutant C43S, C356S (No. 37 and 38). To construct pTK-hPGAP4-3HA, pME-hPGAP4-3HA was cut by SalI and XbaI and the obtained inserts were ligated with pTK-HA-hPIGY cut by XhoI and XbaI. To make the Cas9 expression plasmids, pX330-EGFP was cut by BbsI followed by ligation with annealed primers designated as follows: hamPGAP4#1 (primers No. 1 and 2), hamSLC35A2#2 (No. 3 and 4), hamPIGN#1 (No. 5 and 6), hamPIGN#2 (No. 7 and 8), hamPGAP3#1 (No. 9 and 10), hamPGAP3#2 (No. 11 and 12), and hamPGAP2#1 (No. 13 and 14). Sequences for sgRNAs were designed using CRISPRdirect[42]. The expression plasmid for hPIGN, hamPGAP5, and rPGAP2 were described previously and are available upon request[3, 6, 43].

**Cell lines**. CHO-K1 and Lec8 cells were obtained from ATCC. 3B2A cells were derived from CHO-K1 by stably expressing human DAF and CD59[44]. All cell lines used were cultured in Dulbecco's modified Eagle's medium/F-12 (Nacalai Tesque) supplemented with 10% fetal calf serum at 37 °C under 5% v/v CO$_2$ condition. Cells except Lec8 cells were maintained in a medium supplemented with 600 µg ml$^{-1}$ G418. Cells expressing His-FLAG-GST-FLAG-CD59 (HFGF-CD59) were cultured in a medium supplemented with 600 µg ml$^{-1}$ G418 and 8 µg ml$^{-1}$ blasticidin S. Cells harboring empty vector or expression plasmids except HFGF-CD59 were cultured in a medium supplemented with 600 µg ml$^{-1}$ G418, 8 µg ml$^{-1}$ blasticidin S, and 800 µg ml$^{-1}$ hygromicin B.

To establish the 3BT5 cell line, T5-positive cells were enriched from wild-type 3B2A cells by three rounds of cell sorting, followed by limiting dilution. One clone, termed 3BT5, showed clear T5 staining without a T5 staining-negative population.

To make KO cell lines, pX330-EGFP plasmids harboring sgRNA of target genes were transiently transfected into cells by electroporation as described below. Three days after transfection, cells with EGFP fluorescence were collected by fluorescence-activated cell sorter (FACS) Aria II (BD), followed by limiting dilution to isolate KO clones. At least 1 week later, KO was validated.

To make stable transfectants except for 3BT5-PGAP4-KO cells expressing 3FLAG-hPGAP4(Q26K)-3HA, cells were infected with retrovirus harboring pLIB2-Hyg or pLIB2-BSD with relevant inserts in antibiotic-free medium as described below. To generate 3BT5-PGAP4-KO cells expressing 3FLAG-hPGAP4 (Q26K)-3HA, cells were electroporated with pME-Hyg-3FLAG-hPGAP4-3HA linearized by FspI. After 3 days culture post infection or transfection, cells were selected by the indicated antibiotics.

**Plasmid transfection by electroporation**. Cells ($5 \times 10^6$) were suspended in 400 µl of Opti-MEM and electroporated with 10 µg of plasmid at 260 V and 1000 µF using a Gene Pulser (Bio-Rad). Cells were then cultured for 3 days in antibiotic-free medium.

**Infection of retrovirus**. Cells receiving retrovirus were transiently transfected with pME-mCAT1 by electroporation. On the next day, culture medium was changed followed by further incubation for 2 days. PLAT-E packaging cells (Cell Biolabs) were seeded on 12- or 6-well plates at 70% confluency. On the next day, cells were transfected with pLIB2-BSD or pLIB2-Hyg plasmid bearing cDNA of interest by using PEImax transfection reagent (Polysciences, 24765-2). After about 10 h culture, 1 M sodium butyrate was added to 10 mM final concentration and further incubated for 12 h. On the next day, medium was replaced by a fresh one and incubated at 32 °C for 24 h. Culture medium was added to cells expressing mCAT1, and these cells were cultured at 32 °C for 24 h. On the next day, culture medium was changed, and cells were cultured at 37 °C.

**Gene-trapping screening**. pCMT-SApA-BSD plasmid that contains a splice acceptor sequence for gene-trapping was constructed previously[22]. As recipient cells, 3BT5 cells in one 15-cm dish were transfected with pME-mCAT1 plasmid by electroporation followed by incubation at 37 °C in eight 6-well plates for 3 days. For virus production, PLAT-E cells in two 15-cm dishes were transfected with pCMT-SApA-BSD plasmid using calcium phosphate transfection method. Twelve hours later, 1 M sodium butyrate was added to 10 mM final concentration. Twelve hours later, medium was removed and fresh medium was added followed by incubation at 32 °C. Twenty four hours later, supernatant containing virus was mixed with 8 µg ml$^{-1}$ polybrene. 3BT5 cells expressing mCAT1 were infected with the prepared virus solution by centrifugation at $1220 \times g$ for 2 h at 32 °C. From 2 days after infection, selection was performed with 8 µg ml$^{-1}$ blasticidin S for 1 week. Mutant cells with negative staining with T5 antibody (1:100 ascites) were collected twice by cell sorting using FACS Aria II to enrich mutant cells defective in GPI-GalNAc biosynthesis. For third round of sorting, mutant cells were double stained with T5 and anti-CD59 (10 µg ml$^{-1}$) antibodies, and we collected cells with CD59-positive (GPI-AP-positive) but T5-negative staining to eliminate cells defective in GPI-anchor biosynthesis. Cells with CD59 and T5 double-negative staining were also collected for control experiments.

**Next-generation sequencing**. Genomic DNA was prepared from $2 \times 10^7$ cells using the Wizard Genomic DNA Purification Kit (Promega) according to the manufacturer's protocol. Genomic DNA (15 µg) was fragmented by Covaris S2 (M&S Instruments Inc.) according to 800 bp protocol followed by repairing by T4 DNA polymerase (NEB) and phosphorylation by T4 PNK (NEB). Splinkerette adaptors (No. 65 and 66, see Supplementary Table 1)[45] were ligated. After column purification, the fragments were used for templates of splinkerette PCR using primers No. 67 and 68. The resulting DNA fragments were further amplified by nested PCRs using primers No. 69 and 70, followed by primers No. 71 and 72. Illumina P5 and P7 sequences and barcode sequences were attached to the products by 4 cycles of PCR with 100 ng each of initial PCR product as the template. Paired-end sequencing (101-bp × 2) was performed on the HiSeq 2500 (Illumina) platform. The numbers of reads obtained from control and S3 cells were approximately 15 million and 9 million, respectively.

**Analysis of gene-trap insertions**. In order to map reads derived from next-generation sequencing described in the preceding section, we used the genome assembly from the CHO cell line CHO-K1 (Accession: GCF_000223135.1, Beijing Genomic Institute)[46] as templates. Because the genomic assembly of CHO cells was incomplete and the entire genomic sequences were assembled in 0.1 million scaffolds, we discarded scaffolds <1 Mbp and used only 641 long scaffolds as mapping templates. FASTQ data files were analyzed by the CLC Genomic Workbench software version 7.0.4 (QIAGEN)[21, 22]. In brief, after quality trimming and removal of the common LTR sequence, the 50 base pair reads were mapped onto 641 selected scaffolds. To avoid duplicate mapping, duplicate reads were removed. RPKM score of each scaffold was calculated and ranked. The regions highly mapped by the reads were picked up and the sequences were analyzed by Basic Local Alignment Search Tool (BLAST).

**Flow cytometry**. To detect free GalNAc-GPI, cells were treated with or without final 2 mg ml$^{-1}$ pronase in Opti-MEM containing 1 mM CaCl$_2$ at 37 °C for 2 h. Cells were stained with mouse monoclonal anti-free GalNAc-GPI (T5-4E10) (1:100 ascites) in FACS buffer (phosphate-buffered saline (PBS) containing 1% bovine serum albumin (BSA) and 0.1% NaN$_3$) followed by rat monoclonal anti-mouse immunoglobulin M (IgM) conjugated with APC or goat polyclonal mouse anti-IgM conjugated with FITC. To detect GPI-APs, anti-CD59 (10 µg ml$^{-1}$) and anti-uPAR (10 µg ml$^{-1}$) followed by polyclonal goat anti-mouse IgG conjugated with PE (1:100). Cells were also stained with FLAER (Cedarlane Laboratories, 1:100), which is a FITC-conjugated non-toxic bacterial toxin aerolysin specifically binding to GPI-APs. For lectin staining, lectins were diluted (1:200) in FACS buffer containing

**Fig. 7** Acceptor substrate specificity, timing of GalNAc modification, and T5 mAb epitope characteristics. **a** Schematic of the structure of GPI-APs in various mutant cells. **b** T5 mAb staining of PIGN-KO cells. (Upper) Surface expression of CD59 and the staining with T5 mAb analyzed by FACS. Dotted lines, background staining; blue lines, 3BT5 cells; red lines, PIGN-KO cells. (Lower) Mean fluorescence intensity (±SD) from three independent experiments. Statistical analyses were done by unpaired Student's t-test. *$p < 0.05$. #1 and #2, clones generated by two different sgRNAs. **c** Percentage of GalNAc modification of CD59-GPI in PGAP5 mutant (C19) cells. C19-SLC35A2-KO cells were transiently transfected with empty vector or pME-hyg-hamPGAP5. Percentage of total intensity (mean ± SD) was calculated from the peak areas obtained by two independent measurements with LC-ESI-MS/MS. **d** Percentage of GalNAc modification of CD59-GPI in PGAP2-KO cells. **e** The proposed biosynthetic pathway of GPI-APs in the Golgi. Fatty acid remodeling by PGAP3 and PGAP2 and then GalNAc side-chain modification by PGAP4 occur. GalNAc residue can be further elongated by galactose (Gal) and sialic acid (Sia). GalT and SiaT for GPI modification are not identified yet. UDP-Gal is supplied by SLC35A2. **f** Flow cytometry of PGAP3-KO cells. Staining for T5, CD59, uPAR, and FLAER. Mean fluorescence intensity (±SD) from three independent experiments. Statistical analyses were done by unpaired Student's t-test. *$p < 0.05$; **$p < 0.005$. #1 and #2, clones generated by two different sgRNAs. **g** Percentage of GalNAc modification of CD59-GPI in PGAP3-KO cells. See also Supplementary Fig. 3, 10, 11 and 12

1 mM CaCl$_2$, MgCl$_2$, and MnCl$_2$. The data were collected by FACS Canto II (BD) and analyzed by the FlowJo software.

**Genotyping of 3BT5 cell line**. mRNA of 3BT5 was prepared from $1 \times 10^6$ cells by the RNeasy Mini Kit (QIAGEN). A sample of cDNA was prepared by reverse transcription reaction using SuperScript VILO Master Mix (Thermo Fisher). SLC35A2 was amplified by specific primers (No. 73 and 74) followed by Sanger sequencing.

**DRM separation**. Cells were lysed in lysis-buffer A containing 50 mM Tris-HCl (pH 7.4), 150 mM NaCl, 1% Triton X-100, 5 mM EDTA, and 1× protease inhibitor cocktail followed by centrifugation at $21,900 \times g$ for 15 min at 4 °C. Supernatants were collected as Triton-soluble fraction (S). Pellets containing DRM fraction were suspended by lysis-buffer B containing 50 mM Tris-HCl (pH 7.4), 150 mM NaCl, 60 mM Octyl-β-D-glucoside, 1 mM EDTA, and 1× protease inhibitor cocktail followed by centrifugation at the same condition. Supernatants were collected as Triton-resistant fraction (R). Each of S or R samples was mixed by opposite buffer to set same buffer conditions. 4× sodium dodecyl sulfate (SDS) sample buffer without β-mercaptoethanol was added followed by boiling at 95 °C for 5 min. Samples were then analyzed by western blotting.

**Western blotting**. For the PGAP4 expression analysis, cell lysates were prepared by lysis-buffer A. Lysates were centrifuged at $21,900 \times g$ for 15 min at 4 °C. Supernatant was recovered and mixed with 4× SDS-sample buffer with 5% β-mercaptoethanol followed by incubation on ice for 30 min. Samples were run on SDS-polyacrylamide gel electrophoresis (PAGE) gels and transferred to poly-vinylidene difluoride membranes. Antibodies used were mouse monoclonal anti-HA (1:2000), anti-TfR (1:1000), anti-GAPDH (1:4000), anti-CD59 (0.5 µg ml$^{-1}$), anti-DAF (0.5 µg ml$^{-1}$), anti-uPAR (0.5 µg ml$^{-1}$), and anti-Caveolin 1 (1:1000). Relevant areas in blots are shown in figures and scans of uncropped blots are shown in Supplementary Fig. 13.

**Protein purification**. For purification of HFGF-CD59[6], $2 \times 10^8$ cells stably expressing HFGF-CD59 were treated with 1 unit ml$^{-1}$ PI-PLC in 10 ml of PI-PLC buffer (Opti-MEM containing 10 mM HEPES-NaOH (pH 7.4), 1 mM EDTA and 0.1% BSA) at 37 °C for 2 h. A sample was loaded to a column of 500 µl of Glutathione Sepharose 4B at 4 °C. After washing with 10 ml PBS, HFGF-CD59 was eluted by elution-buffer A (pH7.4, PBS containing 30 mM HEPES-NaOH (pH 7.4) and 20 mM reduced glutathione). Trichloroacetic acid (TCA; Wako) was added to eluted samples to final 10% followed by incubation on ice for 30 min. Proteins were precipitated by centrifugation at $13,400 \times g$ for 30 min at 4 °C. After twice washing with 100% ice-cold ethanol, pellets were resolved in 1× SDS-sample buffer with 5% β-mercaptoethanol and boiled at 95 °C for 5 min. Samples were run on SDS-PAGE gel and stained by Imperial Protein Stain. For PGAP2-KO cells, 200 ml culture medium from ten 15-cm dishes were loaded to a column of 500 µl of Glutathione Sepharose 4B followed by the same procedures described above.

To determine the disulfide bonds of PGAP4, $2 \times 10^8$ cells stably expressing 3FLAG-hPGAP4(Q26K)-3HA were lysed in lysis-buffer A. After centrifugation at $13,400 \times g$ for 15 min at 4 °C, supernatant was loaded to a column of 500 µl of FLAG-M2 beads (Sigma-Aldrich) at 4 °C. After washing with 10 ml PBS containing 1% Triton X-100, 3FLAG-hPGAP4(Q26K)-3HA was eluted by elution-buffer B (PBS containing 20 mM HEPES (pH 7.4) and 150 µg ml$^{-1}$ 3×FLAG peptide). Eluted samples were precipitated by TCA. Pellet was resolved in 1× SDS-sample buffer without β-mercaptoethanol to keep disulfide bridges. Samples were run on SDS-PAGE gel and stained by Imperial Protein Stain.

**Mass spectrometry**. To analyze the GPI-containing peptides from HFGF-CD59, protein bands were excised and reduced with 10 mM dithiothreitol (DTT), followed by alkylation with 55 mM iodoacetamide, and digested in-gel by treatment with trypsin. The resultant peptides were subjected to mass spectrometric analyses by MALDI-TOF or LC-ESI system. MALDI-TOF MS analysis was performed by AXIMA Resonance (Shimadzu/Kratos) in the positive ion mode. The system was controlled by the Shimadzu Biotech MALDI-MS software. Ionization was performed with 337 nm pulsed N$_2$ laser. Helium and argon gases were used for ion cooling and collision-induced dissociation, respectively. For sample preparation, 1 µl aliquots of the peptides were purified with Zip-Tip$_{\mu\text{-}C18}$ pipette tips (Millipore) and deposited directly in matrix (0.5 µl of 5 mg ml$^{-1}$ 2,5-dihydroxybenzoic acid (Shimadzu GLC) in 50% acetonitrile containing 0.05% trifluoroacetic acid) onto the MALDI target plate. The solution on the target plate was completely dried, and then the plate was introduced into the instrument. The instrument was calibrated using an external standard peptide mixture of human ACTH peptide fragment 18–39 ([M + H]$^+$, 2465.20) and bovine insulin oxidized B chain ([M + H]$^+$, 3494.65). For quantification, data were acquired by nanocapillary reversed-phase LC-MS/MS using a C18 column ($0.1 \times 150$ mm) on a nanoLC system (Advance, Michrom BioResources) coupled to an LTQ Orbitrap Velos mass spectrometer (Thermo Fisher). The mobile phase consisted of water containing 0.1% formic acid (solvent A) and acetonitrile (solvent B). Peptides were eluted by a gradient of 5–35% B for 45 min at a flow rate of 500 nl min$^{-1}$. The mass scanning range of the instrument was set at $m/z$ 350–1500. The ion spray voltage was set at 1.8 kV in the positive ion mode. The MS/MS spectra were acquired by automatic switching between MS and MS/MS modes (with collisional energy set to 35%). Helium gas was used as collision gas.

Xcalibur Software (Thermo Fisher) was used for analysis of mass data. In the MS/MS profiles, those that contain characteristic fragments derived from GPI anchors, such as fragment ions of $m/z$ 422$^+$ and 447$^+$, were selected, and fragments in the selected profiles were assigned to determine the GPI structures. Based on the profiles of the MS/MS fragments, the peak areas of the parental MS fragments corresponding to predicted GPI-peptides (Supplementary Table 2, 3, 4, 5) were measured and the ratio in the total GPI peptide fragments was calculated.

To determine the disulfide bridges in PGAP4, bands were excised after SDS-PAGE under non-reducing conditions and were subjected to in-gel digestion by chymotrypsin either without or with reduction and alkylation with 10 mM DTT and 55 mM iodoacetamide (+DTT). Peptides from the digested samples were analyzed by nanoLC-MS/MS. Peptides corresponding to PGAP4 were identified by database searching in-house MASCOT Server (Matrix Science). Precursor mass tolerance was set to 10 ppm and 0.8 Da for Orbitrap and linear ion trap, respectively. Dehydrogenation of cysteine and oxidization of methionine were set as dynamic modification. Disulfide-linked peptide adducts were identified using an algorithm DBond developed by Choi et al.[47] (Supplementary Fig. 7b and c and Supplementary Table 6).

**Immunofluorescence imaging**. The cells were fixed with PBS containing 4% paraformaldehyde for 20 min at room temperature, followed by washing with 40 mM glycine (pH 7.4) for 10 min. After permeabilization with 0.1% saponin or Triton X-100, cells were double stained with mouse anti-HA (1:100) and rabbit anti-GPP130 (1:200) or anti-BiP (1:100) antibody followed by Alexa 488-conjugated goat anti-mouse IgG (1:1000) and Alexa 594-conjugated goat rabbit-mouse IgG (1:1000). Confocal images were acquired on a FluoView FV1000 (Olympus).

**In silico analysis**. PGAP4 transmembrane segments were determined using MEMSAT3[48], TMHMM[49], TMPRED[50], HMMTOP[51], and Phobius[52] tools. Various protein topology prediction tools predicted first TMD (TM1) to be located at the beginning of N-terminus and two consecutive TMDs (TM2 and TM3) are located in the middle of the sequence. A consensus from all the prediction tools was considered and locations of TM1, TM2, and TM3 are predicted to be W20-A42, I262-A284, and S289-V308, respectively. The BLAST tool[53] was used to align PGAP4 sequences against those present in PDB[54]. Levels of PGAP4 mRNA in various tissues were analyzed by BioGPS[23].

**Comparative modeling of PGAP4 structure**. Owing to a lack of sufficient sequence similarity with any known protein in the PDB data bank, homology modeling of the PGAP4 was not straightforward. We first modeled an initial structure of the PGAP4ΔTM2, TM3 (Δ262–308) using a hierarchical protein structure modeling approach of server I-TASSER[28]. I-TASSER successfully identified structural templates from the PDB by multiple threading approach LOMETS[55] and constructed an atomic model of PGAP4ΔTM2, TM3 by iterative template fragment assembly simulations. We selected the top scoring atomic model for analysis. The overall quality of selected homology model analyzed by PROSA-Web[29] shows a poor score (score = −4.89) but it is still within the range of low-resolution experimental structures of this size of 356 residues. The Ramachandran plot analysis of the backbone residues generated by RAMPAGE[30] shows 71.5% residues in the favored region, 17.2% in the allowed region, and 11.3% in the outlier region. Thus our structure model satisfies the quality considering its rather large size, which could be a limitation for modeling. Three disulfide bridges (C132–C136, C144–C194, and C332–C333) confirmed by LC-ESI-MS/MS analysis were feasible in this model. The distances between Cα carbons of C132–C136, C144–C194, and C332–C333 pairs were 10.2 Å, 8.9 Å, and 3.8 Å, respectively. The presence of a disulfide bridge C144–C194 in the PGAP4 provided a solid basis to validate overall topology of the GT-A fold in modeled structure. Out of the top 10 templates selected from the LOMETS threading programs, the structure of bacterial cellulose synthase[27] (PDB ID: 4P02) showed sufficient structural similarity (Template Modeling score = 0.813).

Based on this sequence and structural similarity information from the I-TASSER server, we built a model of PGAP4ΔN using homology modeling approach in MODELLERv17[31]. Since templates identified by the meta-server threading approach implemented in LOMETS (integrated with I-TASSER) finds some significant threading alignment with PDB library, we used X-ray structure of cellulose synthase and fold recognition model of GT-A fold (template structures) together with sequence alignment between PGAP4 and cellulose synthase GT-A fold region obtained from pGenThreader[56] to model PGAP4ΔN structure. Since there was not enough structural similarity of transmembrane regions with transmembrane region of PGAP4 or any other structure in protein databank, we align TM region residues to gaps in the template sequences (Supplementary Fig. 6a, b). We further used secondary structure information and restrains to model the structure and orientation of TMDs.

The secondary structure information predicted by program PSIPREDv3.3[57] was used to restrain residues to achieve proper α helice and β sheet structures. It is

achieved by special restraints on residues W20-L41, R45-F51, L53-N59, Q63-E81, H116-Q131, D190-S197, Q219-F232, M260-R282, W291-L307, A344-S352, and K362-A372 to form α helices and residues I104-V109, Y206-M209 to be β sheets. Additionally, restrains to form disulfide bridges between C132–C136, C144–C194, and C332–C333 pairs were introduced. The secondary structure of TMDs was set to α helix through restraints. The backbone carbon atoms of W20-F283, N63-Y117, E82-V326, A39-Y119, and L53-D110 residues were harmonic restrained to attain a position within 10 Å from each other.

Finally, MODELLER was run using the slow level of VTFM optimization repeated twice for 300 iterations. This thorough setup is slower and time-consuming but expected to result in more accurate model construction. Regions that were not covered by alignment were modeled by loop modeling procedure in MODELLER. The DOPE (Discrete Optimized Protein Energy)[58] and GA341 scores[59] were calculated for each model and the top 10 models were analyzed visually. Top scoring model without any knot was selected as a final model and discussed here. The overall quality of final model analyzed by RAMPAGE shows 88.2% residues in the favored region, 5.5% in the allowed region and 6.3% in the outlier region.

**Quantification and statistical analysis**. Statistical analyses were done using Microsoft Excel. To compare between two individual groups, unpaired Student's t-test was used. The number of repeats of each experiment is indicated at each figure legend. All quantitative data presented here are means ± SD. We considered p-values <0.05 as statistically significant.

**Data availability**. The mass spectrometric proteomics data have been deposited to the ProteomeXchange Consortium via the PRIDE[60] partner repository with the dataset identifiers PXD008137 (project name "Determination of disulfide bridges of human PGAP4 protein") and PXD008230 (project name "Determination of GPI structure of human CD59 in various KO CHO cells"). All other data supporting the findings are available from the corresponding author.

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

## Acknowledgements

We thank Dr. J.F. Dubremetz (Montpellier University) for T5 mAb, Dr. Y. Tashima for discussion, K. Nakamura and Y. Kabumoto for cell sorting, and K. Kinoshita and A. Kawate for technical help. T.H. was supported by grant-in-aid for Japan Society for the Promotion of Science Fellows and for Research Activity start-up. This work was supported by JSPS/MEXT KAKENHI Grant Numbers JP16H04753 and JP17H06422 to T.K. and a grant for Joint Research Project of the Research Institute for Microbial Diseases, Osaka University to Y.Y. and T.K. S.K.M. was supported by the Tokyo Biochemical Research Foundation.

## Author contributions

T.H. performed experiments and analyzed the data. S.K.M. and Y.Y. created the 3D structure model. S.N. and D.M. conducted next-generation sequencing and analyzed the data. K.S. and Y.T. performed mass spectrometry and analyzed the data. M.F. and T.K. developed the research concept and M.F. initiated the research. N.K., Y.M., and Y.M. gave critical discussions. T.H., S.K.M., S.N., K.S., Y.Y., and T.K. wrote the manuscript. All authors contributed to editing the manuscript.

## Additional information

**Competing interests:** The authors declare no competing financial interests.

