## [Peer Review File · Nature Communications]

Reviewers' comments:

Reviewer #1 (Remarks to the Author):

In their manuscript entitled 'A Golgi GPI-N-acetylgalactosamine transferase PGAP4 having tandem transmembrane regions in the catalytic domain' Kinoshita and colleagues identify the enzyme responsible for the first step of GPI side chain glycosylation (i.e., the GPI-N-acetylgalactosamine transferase that they name PGAP4). Through a rich bouquet of different techniques the authors show that PGAP4 catalyzes the addition of N-acetylgalactosamine to GPI backbone, assess the localization of PGAP4, map the PGAP4 catalyzed reaction in the GPI synthetic pathway, and infer PGAP4 molecular structure. In this last effort the authors report a peculiar organization of PGAP4 as compared to other Golgi localized glycosyltransferases in that PGAP4 has a tandem transmembrane regions in its catalytic domain that the authors suggest would orient the PGAP4 active site towards the plane of the membrane where its substrate (i.e., the GPI anchor) is located. This is an impressive, well written, and well-conducted study that will be of interest for a broad scientific audience. I therefore do not have major concerns and support publication in Nature Communications. Nonetheless, for the sake of increasing the significance of this paper I would suggest the authors to investigate (even for a single GPI anchored protein [CD59 or uPAR]) the impact of PGAP4 deletion on GPI-proteins function (signaling, cell migration). While indeed, the establishment of a PGAP4 KO mouse (as suggested by the authors) will possibly clarify the overall physiological significance of PGAP4 and of GPI glycosylation a preliminary statement about the role of GPI-N-acetylgalactosamination in GPI anchored proteins function would make immediately evident the importance of their discovery.

Reviewer #2 (Remarks to the Author):

The subject of this study and the results of this work fit well to this journal since it addresses an important, but specialized subdiscipline and to this field interesting and novel data are contributed. The problem is clearly defined, the approach is appropriate and the used methodology up-to-date. The manuscript is thereby suitable for publication.

Comment: Unfortunately, the manuscript is a bit difficult to read. There is no good balance between details and main issues. The figures are rather crowded.

English grammar could be better.

Careful text revision could improve the legibility considerably.

Reviewer #3 (Remarks to the Author):

In this manuscript, A Golgi GPI-N-acetylgalactosamine transferase PGAP4 having tandem transmembrane regions in the catalytic domain, the authors embarked on a long journey of investigation, starting from a mAb that recognizes the GalNAc sidechain of GPI lacking Glc, to a mutagenesis screen that identified a GPI-GalNAc transferase, and finally to modeling the structure of this enzyme. Generally speaking, the data are of high quality, the figures effective, the writing concise, and the efforts admirable, but here I limit my comments to the mass spec part.

The MS analysis methods employed in this study are reasonable, but not impeccable. For glycopeptide analysis, the MS data shown explain the structure of the glycan, but not the peptide. For example, in Fig. 2b, Fig. S3, and Fig. S9, the MS or MS/MS spectra of a glycopeptide of hCD59

are shown. The glycan side chain was released from GPI by PI-PLC, and the peptide was generated by trypsin digestion. The MS/MS spectra shown are dominated by glycan fragments, but devoid of peptide fragments, so I conclude that only CID was used for structural interrogation. The peptide sequence DLCNFNEQLEN (underscore indicates glycosylation) was deduced from Mw information only. If correct, this is a half-tryptic peptide. A fully tryptic peptide containing the glycosylation site should be DLCNFNEQLENGGTSLSSEK. Please explain why this peptide was not detected. In Fig. S3c, a MS/MS spectrum was attributed to a non-tryptic peptide KDLCNFNEQLEN carrying the indicated glycan, but the evidence is not convincing—no fragment information about the peptide, like the other MS/MS spectra displayed; but unlike the others, many of the fragment ions annotated as glycan parts hardly stood out above noise. I suggest that the authors could try CID/ETD double play, using CID to sequence the glycan and using ETD to sequence the peptide. HCD of different collisional energy is also worthwhile, with a lower collisional energy for the glycan and a higher one for the peptide.

For disulfide bond analysis, the method used cannot detect a disulfide bond directly unless it is within a peptide, such as in the case of C132-C136 or C332-C333. The C144-C194 disulfide bond was deduced from two separate peptides that were identified only after DTT treatment, one containing alkylated C144 and the other containing alkylated C194. There are database search engines such as pLink-SS (Lu S, Nat Meth, 2015) and DBond (Choi S, JPR, 2010) that are specialized for identification of disulfide bonded peptide pairs. The MS data could be re-analyzed using such tools to identify the C144-C194 disulfide bond directly. Similarly, as the method used is blind to disulfide bonds between two Cys residues that are far away in amino acid sequence, if only a portion of the PGAP4 protein contain the C43-C356 disulfide bond (either intra-molecular or inter-molecular), it will not be detected unless a disulfide bond search engine is used.

Reviewer #4 (Remarks to the Author):

The manuscript by Hirata et al. describes the discovery and characterization of a GPI-N-acetylgalactosamine transferase, PGAP4. The authors use a range of techniques, leaving the reader with confidence that the correct gene was found, that it indeed catalyzes the proposed reaction, that the active site is where the authors say it is and that the structural model is good enough to explain the experimental results. The study is well-designed and the conclusions well-supported. I, therefore, recommend Nature Communications to consider this manuscript for publication.

There are a few parts where the methods and results can be presented in a clearer way:

The terminology in the modeling part should be improved. The method employed here is homology modeling or comparative modeling and not ab initio modeling (albeit distance homology modeling). Please also provide the exact sequence of the protein modeled and the alignment to the template (either sequence or structure based). A few examples where the text should be improved/corrected:

- Line 262: I-TASSER is not an ab initio method, but a fold recognition method.
- Line 267: Reliable is too vague, please rewrite the manuscript to reflect that this is a model and hence is a low-resolution representation.
- Line 271: It will always superimpose well as the structure was used as the template
- Line 276: Again, this is a circular argument (creating a model from a template and then arguing that the model is of the same fold because it looks like the template). I would suggest arguing that PGAP4 is a GT-A based on the template detected by I-TASSER.
- Line 325: Please list the distance constraints used.

The experimental data validating the model is not as strong as the authors claim. The disulfides

were not directly observed but inferred based on 'loss-of-signal', and this can have other explanations besides disulfides. Please add the model(s) as supplemental files in addition to the video. At the resolution of this model, only one of the disulfides will be informative as the two others are local in sequence and it would be easy to imagine that these two constraints would be satisfied even in wildly wrong models. The fact that these models 'look' like proteins (Ramachandran etc.) is not necessarily a consequence that the model is correct but that the modeling programs are using this as part of their scoring.

- Line 322: Please remove 'strongly'
- Line 1055: Two of three disulfides not informative.

The sections reporting on the mass spectrometry work can also be improved. In short, I would recommend the authors looking at publishing guidelines from, for example, Molecular and Cellular Proteomics (<http://www.mcponline.org/site/misc/ifora.xhtml>, Editorial Policies, and Guidelines) to get a feeling for what normally is published. In short, the data acquisition (the protocol) needs to be more accurately described to allow other to reproduce. Second, please add supplemental tables to reflect the search results, third to deposit the data on sites like EBI pride (<https://www.ebi.ac.uk/pride/>) to allow others to access the raw data.

Minor points:

The manuscript is missing articles in a few places, and some prepositions are awkward.

The fonts in the figures should be more similar in size Figure 1B is an example where at least four different font sizes are used, from too small to read (numbers on axis), to (in my opinion), too large for the global x-axis.

Line 157: Give a basic description of the RPKM score before using it as not all readers will be familiar with it.

Reviewer #5 (Remarks to the Author):

This a characteristically detailed and meaningful paper from Kinoshita and colleagues.

GPI membrane anchors are of significant interest to eukaryotic cell biologists of all types. It has been known since 1988 that mammals (and *Toxoplasma*) decorate the core structure of the GPI with a carbohydrate side chain of GalNAc, Gal-GalNAc or sialic acid-Gal-GalNAc. However, the key (GalNAc) glycosyltransferase (GT) - so called PGAP4 in this paper - for initiating this sidechain has been elusive.

Taking an inspired approach (using a Mab raised to a related *Toxoplasma* structure - i.e. a GPI core modified with Glc-GalNAc) the authors furnished themselves with a probe for the GPI GalNAc-sidechain in mammalian cells and then cleverly engineered a cell line to give a decent signal by FACS with this antibody so they could use gene-trap technology to find which gene(s) were responsible for forming this epitope. This lead them to PGAP4. The discovery of this gene alone would be sufficient, but it belongs to a new class of 3-tansmbrane domain GTs with similarity to cellulose synthase with the TM domains 'splitting' the conventional GT-A fold. They perform a good and intelligent number of site-directed mutagenesis/complementation studies to support a structural model ("only a model", but it seems to me a good one) that provides a pretty compelling idea of how the GPI acceptor substrate will bind to it.

I am not an expert on the molecular modelling so other referees my comment, but the remainder of the data are extremely high-quality and well interpreted.

A few very minor comments:

1. line 33: 'ensured' is an odd choice of word
2. line 522-523: Pronase can indeed cleave between C-terminal AA and ethanolamine (eg. Almeida et al (2000) EMBO J)
3. line 568-573: the dataset of which GPIs do and do not contain GalNAc side-chains is a bit richer than referenced (eg those that do also include porcine and human membrane dipeptidase, Torpedo AchE and human CD59).

Reviewer #1 (Remarks to the Author):

In their manuscript entitled ‘A Golgi GPI-N-acetylgalactosamine transferase PGAP4 having tandem transmembrane regions in the catalytic domain’ Kinoshita and colleagues identify the enzyme responsible for the first step of GPI side chain glycosylation (i.e., the GPI-N-acetylgalactosamine transferase that they name PGAP4). Through a rich bouquet of different techniques the authors show that PGAP4 catalyzes the addition of N-acetylgalactosamine to GPI backbone, assess the localization of PGAP4, map the PGAP4 catalyzed reaction in the GPI synthetic pathway, and infer PGAP4 molecular structure. In this last effort the authors report a peculiar organization of PGAP4 as compared to other Golgi localized glycosyltransferases in that PGAP4 has a tandem transmembrane regions in its catalytic domain that the authors suggest would orient the PGAP4 active site towards the plane of the membrane where its substrate (i.e., the GPI anchor) is located. This is an impressive, well written, and well-conducted study that will be of interest for a broad scientific audience. I therefore do not have major concerns and support publication in Nature Communications.

Nonetheless, for the sake of increasing the significance of this paper I would suggest the authors to investigate (even for a single GPI anchored protein [CD59 or uPAR]) the impact of PGAP4 deletion on GPI-proteins function (signaling, cell migration). While indeed, the establishment of a PGAP4 KO mouse (as suggested by the authors) will possibly clarify the overall physiological significance of PGAP4 and of GPI glycosylation a preliminary statement about the role of GPI-N-acetylgalactosamination in GPI anchored proteins function would make immediately evident the importance of their discovery.

Thank you for the suggestion. We plan to address the physiological significance of N-acetylgalactosamination of GPI as a major project in the next few years. At this moment we do not have experimental

settings to assess CD59-dependent signaling and uPAR-dependent cell migration. Moreover, there is no preliminary evidence for involvement of GPI side-chain in these events. We feel it risky to choose CD59 or uPAR among 150 or more GPI-APs as targets to ask physiological significance of GPI-N-acetylgalactosamination. Rather we would like to start from PGAP4 knockout mice that are being established in our laboratory. So, we would like not to address physiological significance of PGAP4-mediated side-chain modification of GPI-APs in this manuscript. We hope that the Reviewer appreciate our plan and agree to publication of this manuscript in its current revised form.

Reviewer #2 (Remarks to the Author):

The subject of this study and the results of this work fit well to this journal since it addresses an important, but specialized subdiscipline and to this field interesting and novel data are contributed. The problem is clearly defined, the approach is appropriate and the used methodology up-to-date. The manuscript is thereby suitable for publication.

Comment: Unfortunately, the manuscript is a bit difficult to read. There is no good balance between details and main issues. The figures are rather crowded.

English grammar could be better.

Careful text revision could improve the legibility considerably.

With a help of an expert editor, we revised the text carefully for readability and corrections of grammatical errors. Revisions are highlighted in the text. We also revised the figures (especially Figures 4 and 7) to reduce crowdedness.

Reviewer #3 (Remarks to the Author):

In this manuscript, A Golgi GPI-N-acetylgalactosamine transferase PGAP4 having tandem transmembrane regions in the catalytic domain, the authors embarked on a long journey of investigation, starting from a mAb that recognizes the GalNAc sidechain of GPI lacking Glc, to a mutagenesis screen that identified a GPI-GalNAc transferase, and finally to modeling the structure of this enzyme. Generally speaking, the data are of high quality, the figures effective, the writing concise, and the efforts admirable, but here I limit my comments to the mass spec part.

The MS analysis methods employed in this study are reasonable, but not impeccable. For glycopeptide analysis, the MS data shown explain the structure of the glycan, but not the peptide. For example, in Fig. 2b, Fig. S3, and Fig. S9, the MS or MS/MS spectra of a glycopeptide of hCD59 are shown. The glycan side chain was released from GPI by PI-PLC, and the peptide was generated by trypsin digestion. The MS/MS spectra shown are dominated by glycan fragments, but devoid of peptide fragments, so I conclude that only CID was used for structural interrogation.

For structural analysis of GPI-moiety of tagged-CD59, we first treated the cells by PI-PLC to generate lipid-free soluble tagged-CD59 bearing GPI-glycan. It is then affinity-purified and subjected to in-gel trypsin-digestion and LC-ESI-MSMS. This tagged-CD59 has been used as a model GPI-AP for many years and its peptide part connecting to GPI was well characterized previously by MS/MS (for example in Tashima, Y. et al., 2006. PGAP2 is essential for correct processing and stable expression of GPI-anchored proteins. *Mol. Biol. Cell*, 17:1410). Therefore, we did not specifically analyze peptide part and focused on GPI-glycan.

The peptide sequence DLCNFNEQLEN (underscore indicates glycosylation) was deduced from Mw information only. If correct, this is a half-tryptic peptide. A fully tryptic peptide containing the glycosylation site should be

DLCNFNEQLENGGTSLSSEK. Please explain why this peptide was not detected.

DLCNFNEQLENGGTSLSSEK was not detected because GPI is attached to N at the C-terminus of DLCNFNEQLEN making DLCNFNEQLEN-GPI. During this GPI-attachment reaction (transamidation), the C-terminal peptide consisting of GGTSLSSEK and further C-terminal amino acids is released by cleavage between N and G, and does not exist in the purified mature GPI-AP.

In Fig. S3c, a MS/MS spectrum was attributed to a non-tryptic peptide KDLCNFNEQLEN carrying the indicated glycan, but the evidence is not convincing—no fragment information about the peptide, like the other MS/MS spectra displayed; but unlike the others, many of the fragment ions annotated as glycan parts hardly stood out above noise.

A peptide KDLCNFNEQLEN is a product of trypsin-digestion because the sequence was KKDLCNFNEQLEN-GPI having KK. We often found products after cleavage between two lysines.

I suggest that the authors could try CID/ETD double play, using CID to sequence the glycan and using ETD to sequence the peptide. HCD of different collisional energy is also worthwhile, with a lower collisional energy for the glycan and a higher one for the peptide.

As above, the peptide part was well characterized before and structure of GPI-glycan was the target in this study.

For disulfide bond analysis, the method used cannot detect a disulfide bond directly unless it is within a peptide, such as in the case of C132–C136 or C332–C333. The C144–C194 disulfide bond was deduced from two separate peptides that were identified only after DTT treatment, one containing alkylated C144 and the other containing alkylated C194. There are database search engines such as pLink-SS (Lu S, Nat Meth, 2015) and DBond (Choi S,

JPR, 2010) that are specialized for identification of disulfide bonded peptide pairs. The MS data could be re-analyzed using such tools to identify the C144–C194 disulfide bond directly. Similarly, as the method used is blind to disulfide bonds between two Cys residues that are far away in amino acid sequence, if only a portion of the PGAP4 protein contain the C43–C356 disulfide bond (either intra-molecular or inter-molecular), it will not be detected unless a disulfide bond search engine is used.

Thank you very much for the helpful advice of using a database search engine for identification of disulfide bonded peptide pairs. According to the advice, we used an algorithm DBond by Choi et al and successfully identified a peptide pair containing a disulfide between C144 and C194. A peptide pair connecting between C43 and C356 was not found with DBond. We included these points in the revised manuscript (Results in L312, Methods in L945 and Supplementary Figure 7c, d).

Reviewer #4 (Remarks to the Author):

The manuscript by Hirata et al. describes the discovery and characterization of a GPI–N–acetylgalactosamine transferase, PGAP4. The authors use a range of techniques, leaving the reader with confidence that the correct gene was found, that it indeed catalyzes the proposed reaction, that the active side is where the authors say it is and that the structural model is good enough to explain the experimental results. The study is well–designed and the conclusions well–supported. I, therefore, recommend Nature Communications to consider this manuscript for publication.

There are a few parts where the methods and results can be presented in a clearer way:

The terminology in the modeling part should be improved. The method employed here is homology modeling or comparative modeling and not ab

initio modeling (albeit distance homology modeling). Please also provide the exact sequence of the protein modeled and the alignment to the template (either sequence or structure based).

We included the sequence of PGAP4 Δ TM2, TM3 used to model and its alignment with the template sequence as Supplementary Figure 6.

A few examples where the text should be improved/corrected:

- Line 262: I-TASSER is not an ab initio method, but a fold recognition method.

We thank the reviewer for pointing out this. We corrected it to “fold recognition approach”. (L269)

- Line 267: Reliable is too vague, please rewrite the manuscript to reflect that this is a model and hence is a low-resolution representation.

We understand the reviewer’s point. Sentence has been changed to “The overall quality of the predicted model was analyzed by PROSA-Web and RAMPAGE as described in Methods, which indicated that the model structure is of comparable quality to low resolution experimental structure of the similar size.” (L273-274)

- Line 271: It will always superimpose well as the structure was used as the template

We agree with the reviewer but in this sentence, we wanted to show that the alignment region is in the catalytic domain. To emphasize this, we rephrased the sentence to “The modeled PGAP4 Δ TM2, TM3 aligns with the catalytic domain and the preceding helical region of the cellulose synthase”. (L277-278)

– Line 276: Again, this is a circular argument (creating a model from a template and then arguing that the model is of the same fold because it looks like the template). I would suggest arguing that PGAP4 is a GT-A based on the template detected by I-TASSER.

We thank the reviewer for suggestion. We changed the sentence to “Fold similarities between PGAP4 and GT-A fold of cellulose synthase detected by I-TASSER, a fold recognition program, suggests that PGAP4 contains a GT-A fold separated by tandem TMDs”. (L298-299)

– Line 325: Please list the distance constraints used.

We listed distance restraints in Methods and indicated it in Results. (L1008-1009, L1031-1040 and L374)

The experimental data validating the model is not as strong as the authors claim. The disulfides were not directly observed but inferred based on 'loss-of-signal', and this can have other explanations besides disulfides. Please add the model(s) as supplemental files in addition to the video.

According to an advice of Reviewer #3, we successfully used a search engine for disulfide-linked peptides and directly showed SS linkage between C144 and C194. We added the data in Supplementary Figure 7c. We added the PGAP4 Δ N model as Supplementary Figure 8 and also added a PDB file of the model as Supplementary Data 1.

At the resolution of this model, only one of the disulfides will be informative as the two others are local in sequence and it would be easy to imagine that these two constraints would be satisfied even in wildly wrong models. The fact that these models 'look' like proteins (Ramachandran etc.) is not necessarily a consequence that the model is correct but that the modeling programs are using this as part of their scoring.

– Line 322: Please remove 'strongly'

We removed 'strongly'.

- Line 1055: Two of three disulfides not informative.

We removed two in-sequence disulfides from the point. (L1009-1010)

The sections reporting on the mass spectrometry work can also be improved. In short, I would recommend the authors looking at publishing guidelines from, for example, Molecular and Cellular Proteomics

(<http://www.mcponline.org/site/misc/ifora.xhtml>, Editorial Policies, and Guidelines) to get a feeling for what normally is published. In short, the data acquisition (the protocol) needs to be more accurately described to allow other to reproduce. Second, please add supplemental tables to reflect the search results, third to deposit the data on sites like EBI pride (<https://www.ebi.ac.uk/pride/>) to allow others to access the raw data.

Thank you for pointing out these issues and for the advice of revisions. We addressed these points and made 3 revisions.

1. More accurately described protocol of the data acquisition is included in Methods (page 29).
2. According to the suggestion, we prepared supplemental tables (Supplementary Table 2, 3, 4, 5 and 6).
3. We deposited the data in the EBI pride. Identifiers are PXD008137 for a project name "Determination of disulfide bridges of human PGAP4 protein", and PXD008230 (DOI: 10.6019/PXD008230) for a project name "Determination of GPI structure of human CD59 in various KO CHO cells". Usernames and passwords are reviewer32485@ebi.ac.uk and eUNRTfPN for PXD008137, and reviewer23150@ebi.ac.uk and bp98vmua for PXD008230, respectively.

Minor points:

The manuscript is missing articles in a few places, and some prepositions are awkward.

With a help of an expert editor, we revised the text carefully for readability and corrections of grammatical errors. Revisions are highlighted in the text.

The fonts in the figures should be more similar in size Figure 1B is an example where at least four different font sizes are used, from too small to read (numbers on axis), to (in my opinion), too large for the global x-axis.

We revised all the figures and fixed this problem.

Line 157: Give a basic description of the RPKM score before using it as not all readers will be familiar with it.

We spelled out RPKM as Reads Per Kilobase of exon per Million mapped reads and added a following explanation in page 7: “RPKM score, which corresponds to a density of sequence reads mapped to exons of a gene,”.

Reviewer #5 (Remarks to the Author):

This a characteristically detailed and meaningful paper from Kinoshita and colleagues.

GPI membrane anchors are of significant interest to eukaryotic cell biologists of all types. It has been known since 1988 that mammals (and *Toxoplasma*) decorate the core structure of the GPI with a carbohydrate side chain of GalNAc, Gal-GalNAc or sialic acid-Gal-GalNAc. However, the

key (GalNAc) glycosyltransferase (GT) – so called PGAP4 in this paper – for initiating this sidechain has been elusive.

Taking an inspired approach (using a Mab raised to a related Toxoplasma structure – i.e. a GPI core modified with Glc–GalNAc) the authors furnished themselves with a probe for the GPI GalNAc–sidechain in mammalian cells and then cleverly engineered a cell line to give a decent signal by FACS with this antibody so they could use gene–trap technology to find which gene(s) were responsible for forming this epitope. This lead them to PGAP4. The discovery of this gene alone would be sufficient, but it belongs to a new class of 3–transmembrane domain GTs with similarity to cellulose synthase with the TM domains ‘splitting’ the conventional GT–A fold. They perform a good and intelligent number of site–directed mutagenesis/complementation studies to support a structural model (“only a model”, but it seems to me a good one) that provides a pretty compelling idea of how the GPI acceptor substrate will bind to it.

I am not an expert on the molecular modelling so other referees my comment, but the remainder of the data are extremely high–quality and well interpreted.

A few very minor comments:

1. line 33: ‘ensured’ is an odd choice of word

We changed ‘ensured’ to ‘showed’.

2. line 522–523: Pronase can indeed cleave between C–terminal AA and ethanolamine (eg. Almeida et al (2000) EMBO J)

Thank you for the suggestion. We revised the sentence and cited the suggested reference (ref #34).

3. line 568–573: the dataset of which GPIs do and do not contain GalNAc side-chains is a bit richer than referenced (eg those that do also include porcine and human membrane dipeptidase, Torpedo AchE and human CD59).

Thank you for pointing out this. We added dipeptidases and human CD59 with citations. We did not include Torpedo AchE because we focused on mammalian GPI-APs.

REVIEWERS' COMMENTS:

Reviewer #1 (Remarks to the Author):

I appreciate the commitment of the authors to approach my comments in the future and I recommend this manuscript for publication in Nature Communications

Reviewer #3 (Remarks to the Author):

The authors have addressed all of my concerns. I am happy to recommend publication of this high-quality work.

Reviewer #4 (Remarks to the Author):

The authors present a new and improved manuscript. All my concerns with the first draft were addressed. I therefore recommend Nature Communications to consider this paper for publication.